# PROBING THE LATENT HIERARCHICAL STRUCTURE OF DATA VIA DIFFUSION MODELS

**Antonio Sclocchi**[*]
Institute of Physics, EPFL

**Alessandro Favero**[*]
Institute of Physics, EPFL

**Noam Itzhak Levi**[*]
Institute of Physics, EPFL

**Matthieu Wyart**
Department of Physics and Astronomy, Johns Hopkins [†]

## ABSTRACT

High-dimensional data must be highly structured to be learnable. Although the compositional and hierarchical nature of data is often put forward to explain learnability, quantitative measurements establishing these properties are scarce. Likewise, accessing the latent variables underlying such a data structure remains a challenge. In this work, we show that forward-backward experiments in diffusion-based models, where data is noised and then denoised to generate new samples, are a promising tool to probe the latent structure of data. We predict in simple hierarchical models that, in this process, changes in data occur by correlated chunks, with a length scale that diverges at a noise level where a phase transition is known to take place. Remarkably, we confirm this prediction in both text and image datasets using state-of-the-art diffusion models. Our results show how latent variable changes manifest in the data and establish how to measure these effects in real data using diffusion models.

## 1 INTRODUCTION

Generative artificial intelligence (AI) systems have demonstrated remarkable capabilities in synthesizing data across various modalities, including images (Betker et al., 2023; Rombach et al., 2022) and text (Brown, 2020; Ouyang et al., 2022; Touvron et al., 2023). The underlying reasons behind these achievements remain poorly understood. Indeed, natural data are often high-dimensional and thus generically intractable due to the curse of dimensionality (Luxburg & Bousquet, 2004; Bach, 2017). Hence, to be learnable, the distribution of the data must be highly structured. Characterizing this structure is a fundamental challenge central to any theory of learning.

*Hierarchical compositionality* (Patel et al., 2015; Mossel, 2016; Poggio et al., 2017; Malach & Shalev-Shwartz, 2018; Schmidt-Hieber, 2020; Cagnetta et al., 2024) is a candidate property put forward to rationalize the success of deep architectures. In this view, data can be decomposed into features organized hierarchically. It is well-established that the grammatical structure of most languages is approximately context-free and hierarchical (Chomsky, 2014; Jäger & Rogers, 2012), although it is unclear how well this structure can capture language semantics (Goldberg, 1995; 2015). Likewise, pattern theory (Grenander, 1996) posits that images have a hierarchical structure. In both cases, obtaining quantitative evidence characterizing this hierarchy and building tools to determine the associated latent variables remain a challenge.

Generative *denoising diffusion probabilistic models* (DDPMs) offer a new handle to tackle this challenge, particularly through **forward-backward experiments**, where a controlled level of noise is added to a starting image and then removed to generate a new one (Ho et al., 2020; Sclocchi et al., 2025; Behjoo & Chertkov, 2023). For small amounts of noise, low-level features of the image change (Ho et al., 2020; Sclocchi et al., 2025). Passed a transition point, the class is likely to change (Ambrogioni, 2023; Sclocchi et al., 2025; Biroli et al., 2024; Li & Chen, 2024), but remarkably some of the low-level features of the original image are still retained, as predicted in

---

[*]Co-first authors. Contact: `first_name.last_name@epfl.ch`
[†]*On leave from* EPFL

simple hierarchical models of data structure (Sclocchi et al., 2025). However, the empirical tests in these works were limited to images and did not explore other data modalities. Moreover, the geometrical structure of the changes occurring in such a process is not known.

In this work, we derive the length scale associated with changes occurring in the forward-backward protocol in a synthetic generative model of hierarchical data, and we show experimentally that our predictions hold in both language and image datasets. The synthetic model we consider belongs to the class of probabilistic context-free grammars (Rozenberg & Salomaa, 1997); it is thus defined on a tree graph, and its forward-backward diffusion experiments can be done exactly with a message-passing algorithm. Specifically, our contributions are as follows:

- In the generative model of hierarchically structured data, using a mean-field description of the forward-backward diffusion process, we show theoretically that changes in the tokens are correlated over a length scale that diverges at the class transition. This phenomenology is a signature of the hierarchy in the data structure, indicating changes in deep latent variables.
- We validate our theoretical predictions performing numerical experiments on our synthetic data with a diffusion process used in practice for discrete data, showing the same phenomenology predicted by our theory. To do so, we measure the *dynamical susceptibility*, an observable used to study the dynamics in physical systems.
- We perform forward-backward experiments with state-of-the-art masked diffusion language models (MDLM) (Sahoo et al., 2024) on `WikiText`. We show the presence of a peaking correlation length in the token changes at a finite inversion time, consistently with our theoretical model.
- We perform the same experiments with vision Denoising Diffusion Probabilistic Models (DDPM) (Nichol & Dhariwal, 2021) on `ImageNet`. We tokenize the resulting images using the patch embeddings of a contrastively pre-trained vision encoder (Radford et al., 2021) and show that the correlations of token changes display a qualitative agreement with our analysis.

Overall, our results show how changes in latent variables affect visible data, and directly support the idea that a hierarchical latent structure is central to both language and vision modalities. Moreover, our work puts forward the forward-backward protocol as a tool to probe the latent hierarchical structure of real data.

**Organization of the manuscript** In Section 2, we provide background on continuous and discrete diffusion models. We then define the theoretical model of hierarchical data we consider and its Bayes-optimal denoising. In Section 3, we define the correlations between changes and associated susceptibility and then present our theoretical result on the hierarchical data model and its experimental validation. In Section 4, we report the experimental measures for the same quantities in language and image diffusion models.

## 2 BACKGROUND

### 2.1 DIFFUSION MODELS

Denoising diffusion models are generative models designed to sample from a distribution by reversing a step-by-step noise addition process (Sohl-Dickstein et al., 2015; Ho et al., 2020; Song & Ermon, 2019; Song et al., 2020). Let $t$ indicate the time step in a sequence $[0, \ldots, T]$, $q(\cdot)$ the data distribution we wish to sample from, and $\boldsymbol{x}_0 \sim q(\boldsymbol{x}_0)$ a sample drawn from this distribution. Diffusion models consist of: a *forward process* generating a sequence of increasingly noised data $\{\boldsymbol{x}_t\}_{1 \leq t \leq T}$, $q(\boldsymbol{x}_1, \ldots, \boldsymbol{x}_T | \boldsymbol{x}_0) = \prod_{t=1}^{T} q(\boldsymbol{x}_t | \boldsymbol{x}_{t-1})$, where at the final time $T$, $\boldsymbol{x}_T$ corresponds to pure noise; a *backward process*, which reverts the forward one by gradually removing noise. This process is typically obtained by learning the backward transition kernels $p(\boldsymbol{x}_{t-1} | \boldsymbol{x}_t)$ using a neural network. This corresponds to learning the *score function*, which is proportional to the conditional expectation $\mathbb{E}_{q(\boldsymbol{x}_0 | \boldsymbol{x}_t)} [\boldsymbol{x}_0]$. Sampling from $q(\cdot)$ is achieved by sampling noise $\boldsymbol{x}_T \sim q(\boldsymbol{x}_T)$ and then applying the learned backward process to obtain a new sample $\boldsymbol{x}_0 \sim q(\boldsymbol{x}_0)$. Different diffusion models correspond to different choices of the forward process, depending on the data space under consideration (see Yang et al. (2023) for a review).

**Discrete data**  For discrete data, like text, $\boldsymbol{x}_0$ consists of a sequence of tokens $x_{0,i}$, $i \in [d]$, each corresponding to a symbol belonging to a vocabulary $\mathbb{V}$. In this case, we consider *masked diffusion with an absorbing state* by introducing an additional [MASK] symbol (Austin et al., 2021). At time step $t$, each non-masked token either stays unchanged or transitions to [MASK] with some probability $\beta_t$. Using a one-hot-encoding representation of these $|\mathbb{V}| + 1$ states, the forward transition matrix reads $q(x_{t,i}|x_{t-1,i}) = (1 - \beta_t)\mathbb{I} + \beta_t \mathbf{1}\mathbf{e}_M^\top$, where $\mathbb{I}$ is the identity matrix, $\mathbf{1}$ a column vector of all ones and $\mathbf{e}_M$ the one-hot-encoding of the [MASK] symbol. At the final time $T$, $x_{T,i} = $ [MASK] for every $i \in [d]$. In the following, we consider the noise schedule $\beta_t = (T - t + 1)^{-1}$ such that $q(x_{t,i} = $ [MASK] $| \boldsymbol{x}_0) = t/T$ (Austin et al., 2021).

**Continuous data**  For continuous data, like images, corresponding to $\boldsymbol{x}_0 \in \mathbb{R}^d$, we consider the time-discretized Gaussian diffusion introduced in (Ho et al., 2020). The forward transition matrix reads $q(\boldsymbol{x}_t|\boldsymbol{x}_{t-1}) = \mathcal{N}(\boldsymbol{x}_t; \sqrt{1 - \beta_t}\boldsymbol{x}_{t-1}, \beta_t\mathbb{I})$, where $\mathcal{N}$ represents the Gaussian probability distribution and the sequence $\{\beta_t\}_{1 \leq t \leq T}$ is the variance schedule. At the final time $T$, $\boldsymbol{x}_T \sim \mathcal{N}(0, \mathbb{I})$.

**Forward-backward experiments**  Forward-backward experiments involve inverting the diffusion process at an intermediate time $t \leq T$. Starting from $\boldsymbol{x}_0$, the forward process up to time $t$ produces a noisy sample $\boldsymbol{x}_t \sim q(\boldsymbol{x}_t|\boldsymbol{x}_0)$. The backward process, obtained from a diffusion model, produces a new sample $\hat{\boldsymbol{x}}_0(t) \sim p(\hat{\boldsymbol{x}}_0|\boldsymbol{x}_t)$. We refer to $t$ as the *inversion time* of the forward-backward process.

## 2.2 THE RANDOM HIERARCHY MODEL (RHM)

The RHM is a generative model of hierarchically structured data introduced by (Cagnetta et al., 2024). It belongs to the class of context-free grammars in the field of language theory (Rozenberg & Salomaa, 1997), and assumes that production rules are random. In its simplest version, it consists of:

- A regular tree graph of depth $L$ and branching factor $s$. Each node of the tree corresponds to a discrete random variable.
- A discrete vocabulary $\mathbb{V}^{(\ell)}$ of size $v$ for each level $\ell = 0, 1, \ldots, L$ of the tree. We call $\ell = 0$ the level of the leaves and $\ell = L$ the root level.
- A set of production rules defining how each symbol belonging to $\mathbb{V}^{(\ell)}$ can be represented at the lower level with the symbols of $(\mathbb{V}^{(\ell-1)})^{\otimes s}$. For each element of $\mathbb{V}^{(\ell)}$, there are $m$ equivalent lower-level representations, which are all distinct and chosen randomly.

We use the notation $\mathrm{h}_i^{(\ell)}$ to indicate the variable at level $\ell$ and position $i \in [s^{L-\ell}]$. The leaf nodes $\mathrm{h}_1^{(0)}, \ldots, \mathrm{h}_{s^L}^{(0)}$ correspond to the visible tokens, while the upper-level nodes represent latent variables. See App. A for an example. We define the tree distance $\tilde{\ell}$ between two visible tokens as the number of edges between them and their lowest common ancestor. Their corresponding real space distance $r$ is $r = s^{\tilde{\ell}} - 1$. Because of the hierarchical structure generating the data, the visible tokens have non-trivial spatial correlations, which depend on their tree distance (Cagnetta & Wyart, 2025).

### 2.2.1 BAYES-OPTIMAL DENOISING OF THE RHM USING BELIEF PROPAGATION

Knowing the production rules and the tree structure of the RHM, the probabilities of the latent variables, conditioned on some observation, can be reconstructed exactly (Sclocchi et al., 2025) using the **Belief Propagation (BP)** algorithm (Mezard & Montanari, 2009). Specifically, if an RHM datum $\boldsymbol{x}_0$ is corrupted by some noise, e.g., via masking a fraction of tokens, resulting in a noisy observation $\boldsymbol{x}_t$, then BP can be used to:

- compute the marginal probabilities of any latent or visible variable, conditioned on the noisy observation $\boldsymbol{x}_t$: $p(\mathrm{h}_i^{(\ell)}|\boldsymbol{x}_t)$;
- sample directly from the posterior $p(\hat{\boldsymbol{x}}_0|\boldsymbol{x}_t)$.

If the noisy observation $\boldsymbol{x}_t$ is produced by a forward diffusion process, then sampling from $p(\hat{\boldsymbol{x}}_0|\boldsymbol{x}_t)$ is equivalent to integrating exactly (i.e., for an infinite number of time steps) the backward diffusion process starting from $\boldsymbol{x}_t$ and using the exact score function. In fact, BP can also be used to compute the score function, which is proportional to $\mathbb{E}_{p(\hat{\boldsymbol{x}}_0|\boldsymbol{x}_t)}[\hat{\boldsymbol{x}}_0]$, corresponding to having access to a neural network achieving perfect generalization (see App. A.1.3 for a comparison between BP sampling and backward diffusion with the score function). This is a different situation with respect to real data, like images and text, where the score is estimated by training a neural network.

### 2.2.2 DIFFUSION PROCESSES IN THE RHM

For the RHM data, we consider two different processes.

- $\epsilon$-**process**   This is a simplified process where one considers any datum $\boldsymbol{x}_0$ that can be generated by the RHM, and assumes that there is some level of uncertainty on each visible token (see App. A.1.2 for details). One then uses BP to compute the probability that the true initial datum was $\hat{\boldsymbol{x}}_0$. The noising process is controlled by a noise-to-signal ratio $\epsilon \in [0, 1]$, which plays the role of time in the standard diffusion processes, such that $\epsilon = 0$ at $t = 0$ and $\epsilon = 1$ at $t = T$. Starting from an RHM datum $\boldsymbol{x}_0$, we indicate with $\boldsymbol{x}_\epsilon$ the noisy observation at the leaf priors. Therefore, BP computes the marginals $p(\mathrm{h}_i^{(\ell)}|\boldsymbol{x}_\epsilon)$ and samples from $p(\hat{\boldsymbol{x}}_0|\boldsymbol{x}_\epsilon)$. We study theoretically this process in section 3.1 through a mean-field approximation, neglecting some fluctuations of the marginal probabilities and averaging over the disorder of the RHM.
- **Masking diffusion with an absorbing state**   This is the diffusion process described in section 2.1 for discrete data, which is commonly used in practice. We study it numerically with BP in section 3.2.

**Phase transition in the class reconstruction of the RHM**     Sclocchi et al. (2025) showed that there exists a regime of the RHM parameters where the probability of reconstructing the class in the $\epsilon$ diffusion process, that is $p(\mathrm{h}_1^{(L)}|\boldsymbol{x}_\epsilon)$, undergoes a sharp phase transition at a critical noise level $\epsilon^*$ in the limit of large $L$. Therefore, sampling $\hat{\boldsymbol{x}}_0(\epsilon) \sim p(\hat{\boldsymbol{x}}_0|\boldsymbol{x}_\epsilon)$, for $\epsilon < \epsilon^*$, $\hat{\boldsymbol{x}}_0(\epsilon)$ and $\boldsymbol{x}_0$ share the same latent $\mathrm{h}_1^{(L)}$ (i.e. they belong to the same class), while, for $\epsilon > \epsilon^*$, the probability that $\hat{\boldsymbol{x}}_0(\epsilon)$ and $\boldsymbol{x}_0$ share the same class corresponds to the random chance $1/v$.

In Fig. 10, we show numerically that also in masking diffusion the probability of reconstructing the class $p(\mathrm{h}_1^{(L)}|\boldsymbol{x}_t)$ undergoes a phase transition at a specific inversion time $t^*$.

## 3   HIERARCHICAL STRUCTURES INDUCE CORRELATED BLOCKS OF TOKEN CHANGES

In this section, we characterize the statistics of how the input tokens change in the forward-backward experiments. Let $x_{0,i}$ denote the $i$-th input token, $i \in [d]$, and $\hat{x}_{0,i}(t)$ the same token after undergoing a forward-backward experiment with inversion time $t$. We seek to compute the correlations between changes in the tokens as a function of the inversion time $t$. For each token position $i$, we introduce a variable $\sigma_i(t)$ characterizing the dynamics.

**Definition 1** (Token change). *If the tokens $x_{0,i}$ and $\hat{x}_{0,i}(t)$ take values in a discrete vocabulary, then $\sigma_i(t)$ is a spin variable defined as*

$$\sigma_i(t) = \begin{cases} +1, & \text{if } x_{0,i} \neq \hat{x}_{0,i}(t), \\ -1, & \text{if } x_{0,i} = \hat{x}_{0,i}(t). \end{cases} \tag{1}$$

**Definition 2** (Dynamical correlation function). *Given the $\sigma_i(t)$ defined above, the dynamical correlation function between the changes of tokens at positions $i$ and $j$, relative to the initial point $\boldsymbol{x}_0$, is defined as*

$$\mathcal{C}_{\boldsymbol{x}_0,ij}(t) = \langle \sigma_i(t)\sigma_j(t) \rangle - \langle \sigma_i(t) \rangle \langle \sigma_j(t) \rangle, \tag{2}$$

*where $\langle \cdot \rangle$ denotes averaging over different stochastic trajectories. The **average dynamical correlation function** is defined as $\mathcal{C}_{ij}(t) = \overline{\mathcal{C}_{\boldsymbol{x}_0,ij}(t)}$, where the overline indicates averaging over the initial point $\boldsymbol{x}_0$.*

Given the correlations, we compute the *dynamical susceptibility* $\chi(t)$, a quantity used to study the dynamics in physical systems (Donati et al., 2002; Toninelli et al., 2005).

**Definition 3** (Dynamical susceptibility). *Given the average correlation function $\mathcal{C}_{ij}(t)$ of Definition 2, the dynamical susceptibility is defined as*

$$\chi(t) = \frac{\sum_{i=1}^d \sum_{j=1}^d \mathcal{C}_{ij}(t)}{\sum_{i=1}^d \mathcal{C}_{ii}(t)}, \tag{3}$$

*where we normalized by the sum of auto-correlations.*

Intuitively, the susceptibility measures the volume of the blocks of tokens that change together.

In the case of the $\epsilon$-process for the RHM, where $\hat{x}_0(\epsilon)$ is sampled from $p(\hat{x}_0|x_\epsilon)$, the same definitions hold for the quantities $\mathcal{C}_{ij}(\epsilon)$ and $\chi(\epsilon)$. In the case of continuous embeddings, where the tokens $x_{0,i}$ and $\hat{x}_{0,i}(t)$ are continuous vectors (see Section 4 for image diffusion), the same definitions for $\mathcal{C}_{ij}(t)$ and $\chi(t)$ hold by redefining $\sigma_i(t)$ as $\sigma_i(t) = \|x_{0,i} - \hat{x}_{0,i}(t)\|$.

## 3.1 MEAN-FIELD THEORY OF THE $\epsilon$-PROCESS OF THE RHM

The average correlation function $\mathcal{C}_{ij}(\epsilon)$ can be computed for the $\epsilon$-process of the RHM through a mean-field approximation. This mean-field approach consists of computing the average BP messages at each layer $\ell$, where the average is performed over the possible realizations of the RHM rules. Let's consider two leaf nodes $h_i^{(0)}$ and $h_j^{(0)}$ connected to the common ancestor $h_k^{(\ell)}$ at layer $\ell$ through the nodes $h_l^{(\ell-1)}$ and $h_m^{(\ell-1)}$ (see Fig. 1 for an illustration). Their associated spin variables are therefore $\sigma_i^{(0)}$, $\sigma_j^{(0)}$, $\sigma_k^{(\ell)}$, $\sigma_l^{(\ell-1)}$ and $\sigma_m^{(\ell-1)}$, where we omit the $\epsilon$ dependence to lighten the notation. Given the tree structure, the joint probability distribution $P(\sigma_i^{(0)}, \sigma_j^{(0)})$ can be written as

$$P(\sigma_i^{(0)}, \sigma_j^{(0)}) = \sum_{\sigma_l^{(\ell-1)}, \sigma_m^{(\ell-1)}} P(\sigma_i^{(0)}|\sigma_l^{(\ell-1)}) \, P(\sigma_j^{(0)}|\sigma_m^{(\ell-1)}) \, P(\sigma_l^{(\ell-1)}, \sigma_m^{(\ell-1)}). \tag{4}$$

Each element in the sum of Eq. (4) can be written in terms of BP messages, and its average value can be computed by averaging over the realizations of RHM rules. The average of $P(\sigma_i^{(0)}|\sigma_l^{(\ell-1)})$ and $P(\sigma_j^{(0)}|\sigma_m^{(\ell-1)})$ can be written as a $2 \times 2$ matrix $T^{(\ell-1)}$ only depending on the layer $\ell - 1$. Similarly, also the average of the joint probability $P(\sigma_l^{(\ell-1)}, \sigma_m^{(\ell-1)})$ can be represented as a $2 \times 2$ matrix $C^{(\ell-1)}$. In the mean-field approximation, we neglect the fluctuations of these quantities around their means. Therefore, we compute the average joint probability $P(\sigma_i^{(0)}, \sigma_j^{(0)})$ by substituting the elements in the product of Eq. (4) with their means. For spin variables $i$ and $j$ at tree distance $\ell$, we have

$$P(\sigma_i^{(0)}, \sigma_j^{(0)}) = T^{(\ell-1)} \, C^{(\ell-1)} \, T^{(\ell-1)^\top}. \tag{5}$$

Figure 1: Example of leaf nodes $h_i^{(0)}$, $h_j^{(0)}$ connected to the common ancestor $h_k^{(\ell)}$ through $h_l^{(\ell-1)}$ and $h_m^{(\ell-1)}$.

All the expressions for the above quantities are reported in App. A.2. With a similar procedure, we can compute the average marginal probability $p(\sigma^{(0)})$. From these quantities, we obtain the average correlation function $\mathcal{C}_{ij}(\epsilon)$ at each noise level $\epsilon$.

### 3.1.1 DYNAMICAL CORRELATION LENGTH

In what follows, we present our main result predicting a power law divergence of the dynamical correlation length at the phase transition. Technical details of the derivation are reported in App. A.2.1.

**Main result** (Divergence of the correlation length). *Consider the RHM in the limit $L \to \infty$, with parameters $v, m, s$ such that the probability of reconstructing the class in the $\epsilon$-process undergoes a phase transition at noise level $\epsilon^*$ (condition 28 in App. A.2). Then, the correlation length $\xi(\epsilon)$ associated to the dynamical correlation function $\mathcal{C}_{ij}(\epsilon)$ diverges at the class phase transition $\epsilon^*$ as*

$$\xi \sim |\epsilon - \epsilon^*|^{-\nu}, \tag{6}$$

*where $\nu$ is a function of $v, m, s$ reported in Eq. (54) of App. A.2.1.*

This divergence of the correlation length at the class transition indicates that large blocks of tokens change in concert. In fact, these large correlated changes are caused by the modifications of deeper and deeper latent variables near the transition (see Fig. 3 for an illustration). At both smaller and larger noise levels, the correlation length decays. This behavior of the dynamical correlation functions implies that the dynamical susceptibility also peaks at the transition, a hallmark of criticality.

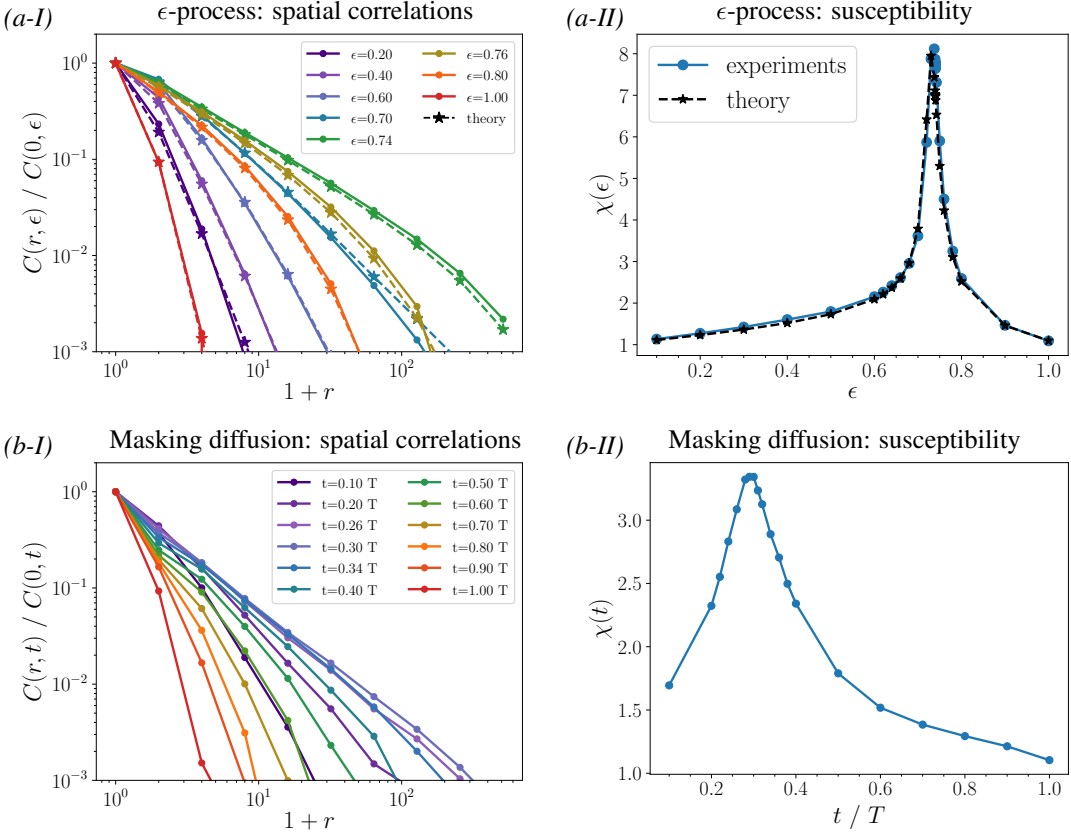

Figure 2: **Correlation measures on diffusion samples of the Random Hierarchy Model (RHM).**
*(a-I)* In the $\epsilon$-process, the average correlation function shows a correlation length that is maximal for $\epsilon^* \simeq 0.74$, corresponding to the class phase transition, with a system-spanning power-law behavior. The full lines are experiments run with Belief Propagation, while the dashed lines are the corresponding mean-field theory description (Section 3.1), showing excellent agreement. *(a-II)* Correspondingly, also the average susceptibility shows a peak at the transition $\epsilon^*$. *(b)* The same behavior is observed for the correlation function *(b-I)* and the susceptibility *(b-II)* for masking diffusion. In this case, the phase transition is observed for inversion time $t^* \simeq 0.3\ T$, where both the correlation length and the susceptibility peak. Data for RHM parameters $v = 32$, $m = 8$, $s = 2$, $L = 9$, averaged over 256 starting data and 256 diffusion trajectories per starting datum.

### 3.2 NUMERICAL EXPERIMENTS

To test our theoretical predictions for the $\epsilon$-process, in Fig. 2 (a-I), we present the average correlation functions $\mathcal{C}(r, \epsilon)$, corresponding to $\mathcal{C}_{ij}(\epsilon)$ averaged on all pairs $ij$ such that their real space distance is $r$, and normalized by the auto-correlation $\mathcal{C}(0, \epsilon)$. We observe that the correlation function displays a system-spanning power-law behavior at a critical value $\epsilon^* \approx 0.74$, while it decays faster with distance when $\epsilon \neq \epsilon^*$. This observation implies a correlation length that peaks at the critical value $\epsilon^*$. Consistently, also the susceptibility $\chi(\epsilon)$ in Fig. 2 (a-II) peaks at this critical value. We compare both the correlation functions and the susceptibility measures with the theoretical predictions obtained by the mean-field theory of the $\epsilon$-process (dashed lines in Fig. 2 (a-I) and (a-II)), showing excellent agreement. Moreover, in Fig. 9 of App. A, we test the prediction for the critical exponent of the correlation length of Eq. (6), also showing very good agreement.

In the panels (b-I) and (b-II) of Fig. 2, we report the average correlation functions $\mathcal{C}(r, t)$ and susceptibility $\chi(t)$ for masking diffusion at different inversion times $t$. Also in this case, the correlation length and the susceptibility are maximal at a specific critical time $t^* \approx 0.3\ T$. From Fig. 10, we observe that this critical time $t^*$ corresponds to the phase transition in the class reconstruction probability. Although there is not a simple mapping between the masking probability $t/T$ and the noise level $\epsilon$ in the simplified $\epsilon$-process, the qualitative behaviors in the two settings show a remarkable agreement, validating the relevance of our theoretical predictions for both kinds of diffusion process.

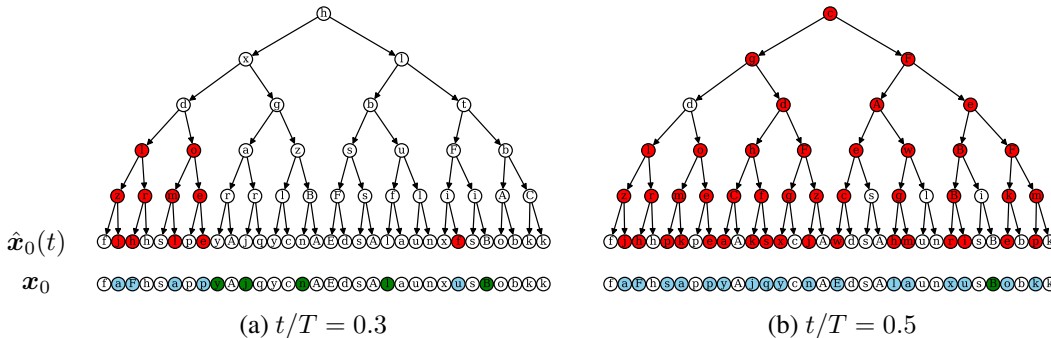

(a) $t/T = 0.3$          (b) $t/T = 0.5$

Figure 3: **Masking diffusion in the RHM for masking fraction** *(a)* $t/T = 0.3$ **and** *(b)* $t/T = 0.5$. The bottom sequence represents the starting datum $\boldsymbol{x}_0$. The blue (green) symbols are the masked ones in $\boldsymbol{x}_t$ that (do not) change feature in $\hat{\boldsymbol{x}}_0(t)$. The leaves of the tree represent the sampled sequence $\hat{\boldsymbol{x}}_0(t) \sim p(\hat{\boldsymbol{x}}_0|\boldsymbol{x}_t)$. In the corresponding tree, the red nodes are those that changed features with respect to the generating tree of $\boldsymbol{x}_0$. We observe that larger blocks of changed tokens reflect changes in deeper latent variables.

### 3.3 Spatial correlations in data are not sufficient to get a susceptibility peak

In the RHM, the latent hierarchical structure induces spatial correlations both between the input tokens and in their changes in the forward-backward diffusion. Therefore, it is natural to ask whether any model of data displaying spatial correlations, even without a latent hierarchical structure, exhibits the same phenomenology of the RHM in the forward-backward diffusion.

In App. B, we show that this is not the case. In particular, we consider a Gaussian random field model with a covariance having a power-law decaying spectrum. This induces spatial correlations in the field that decay algebraically with the distance. We show that performing forward-backward diffusion at different inversion times $t$ results in a variation field having a correlation length that increases monotonically with $t$ and is maximal at the final time $t = T$. This behavior contrasts sharply with the hierarchical data studied here, where the growing length scale occurs in correspondence with a phase transition at a finite inversion time $t^*$.

In fact, the mechanisms behind the dynamical correlations are different. For Gaussian random fields, the noise of the diffusion acts as a low-pass filter, which defines a characteristic scale below which a mode is reconstructed in the backward process. For hierarchically structured data, instead, the spatial correlations in the changes are associated with the changes of latent variables at different levels of the hierarchy. Therefore, a diverging correlation length is present only when the reconstruction probability of the root node (i.e., the class) undergoes a phase transition.

## 4 Experiments on Natural Language and Image Data

This section extends our analysis to real-world scenarios, demonstrating that language and vision diffusion models exhibit the same phenomenology as observed in the RHM.

**Language diffusion models** We consider Masked Diffusion Language Models (MDLM) (Sahoo et al., 2024) utilizing the `GPT2` tokenizer. We perform forward-backward experiments starting from samples from the `WikiText2` dataset. In Fig. 4 (a), we illustrate how an initial paragraph changes with different inversion times $t$. At small $t$, only a few isolated words are modified. At intermediate $t$, we clearly observe clusters of words changing in a correlated manner. At large $t$, only a small fraction of the initial sentence remains unchanged (see App. C for a presentation of the same data in their larger context). In Fig. 4 (b-c), we quantify these observations by measuring the average correlation functions and susceptibility[1]. Strikingly, in line with the phenomenology obtained for the RHM, we find a growing correlation length as $t$ increases, reaching a maximum of $7 \div 8$

---

[1]To avoid finite size effects due to imposing a fixed masking fraction, we integrate the average correlation function up to the maximal correlation length $r \sim \mathcal{O}(10)$.

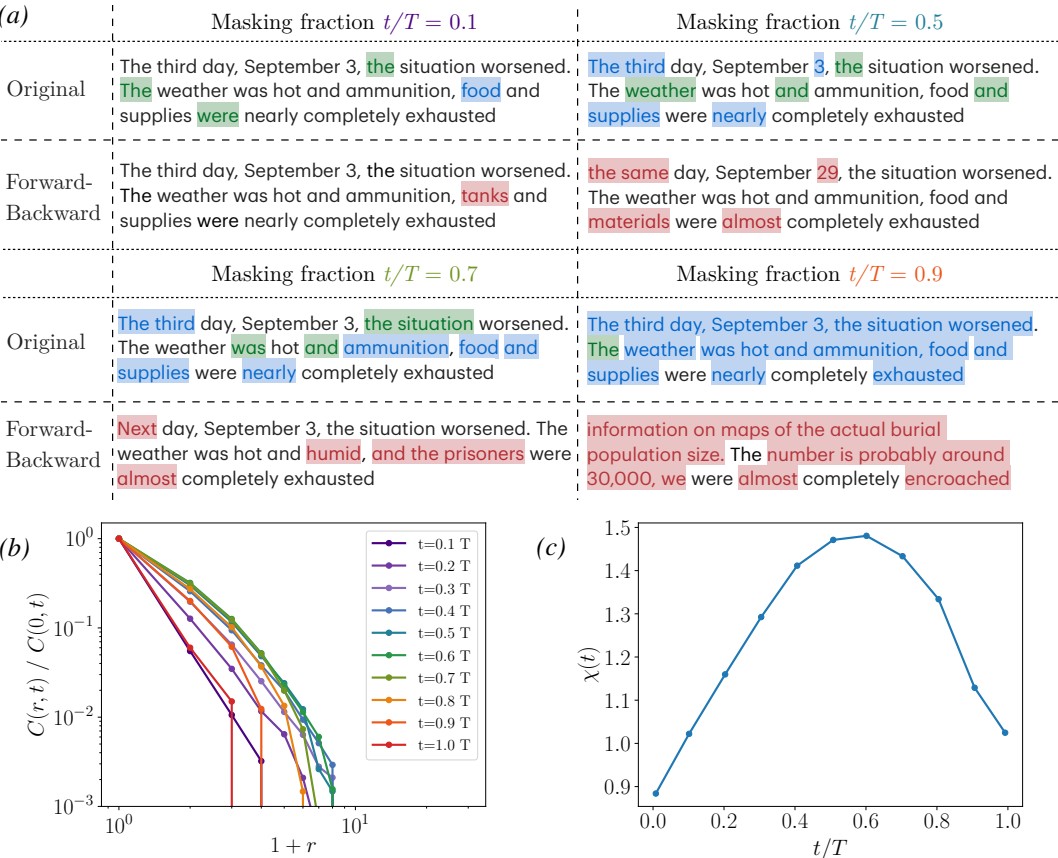

Figure 4: **Forward-backward experiments with language diffusion models.** *(a)* Forward-backward examples for different masking fractions. The words in blue (green) are those that were masked and changed (did not change), while the words in red changed following the backward process. *(b)* Normalized correlations as a function of index distance $r = |i - j|$ for different fractions of masked tokens. *(c)* Susceptibility $\chi(t)$ as a function of masking fraction. The results are averaged over $N_S = 300$ samples, each consisting of $N_T = 128$ tokens, with $N_R = 50$ noise realizations for each masking fraction. The susceptibility is given by integrating over the domain $r \in [0, 10]$.

tokens at a critical inversion time $t^* \approx 0.6\,T$, followed by a subsequent decline. Additionally, the susceptibility peaks at $t^*$, establishing the existence of a phase transition for the language modality.

**Vision diffusion models** We extend our analysis to computer vision by considering Improved Denoising Diffusion Probabilistic Models (Nichol & Dhariwal, 2021), trained on the `ImageNet` dataset. To compute the correlation between changes in the image tokens, we follow recent trends in multimodal LLMs (Liu et al., 2024; Dai et al., 2023). Specifically, we divide each image into $7 \times 7$ patches and use the last-layer embeddings for each patch from a CLIP ViT-B32 (Radford et al., 2021) to tokenize the image. Let $\boldsymbol{x}_{0,i}$ denote the embedding of the $i$-th patch, where $i = (k, l)$ with $k, l \in [7]$. After the forward-backward process, the variation of each patch embedding is given by $\Delta\boldsymbol{x}_i(t) = \boldsymbol{x}_{0,i} - \hat{\boldsymbol{x}}_{0,i}(t)$. We then compute the average correlations between the norms of these variations:

$$\mathcal{C}_{ij}(t) = \overline{\langle \|\Delta\boldsymbol{x}_i(t)\| \, \|\Delta\boldsymbol{x}_j(t)\| \rangle - \langle \|\Delta\boldsymbol{x}_i(t)\| \rangle \langle \|\Delta\boldsymbol{x}_j(t)\| \rangle}. \tag{7}$$

The susceptibility is subsequently obtained as $\chi(t) = \sum_{ij} \mathcal{C}_{ij}(t) / \sum_{ii} \mathcal{C}_{ii}(t)$. In Fig. 5, we present some examples of starting images and generated ones at different inversion times $t$, together with the grid representing their tokenization. We observe that, for increasing $t$, new semantic elements appear in the generated images, corresponding to blocks of tokens changing in concert. In Fig. 6, we present the average correlation functions and the susceptibility for vision DDPMs, starting from samples of the `ImageNet` validation set (Deng et al., 2009). At a critical inversion time $t^* \approx 0.6 \div 0.7\,T$, we observe a peak in susceptibility, signaling the class phase transition in these

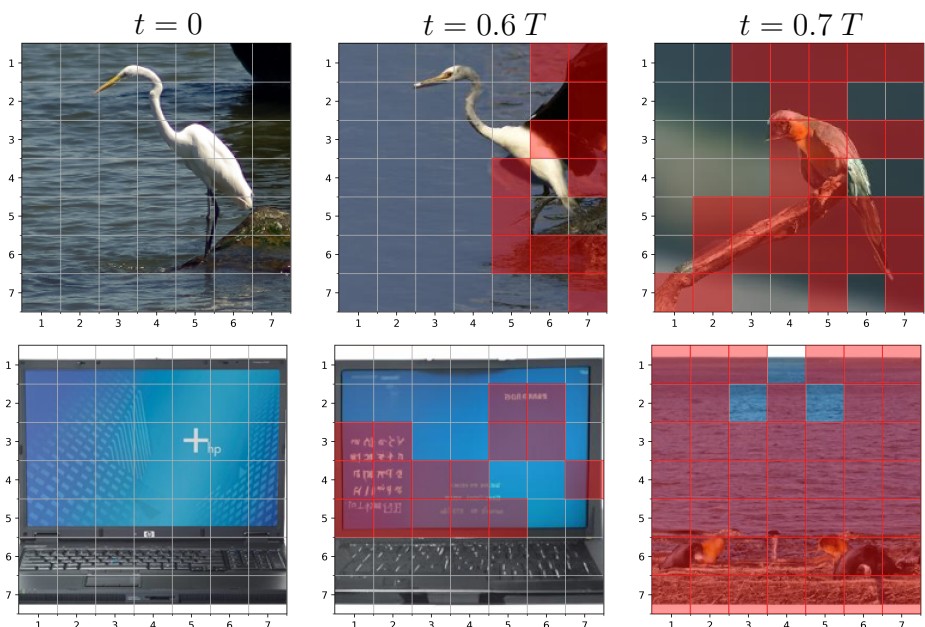

Figure 5: **Examples of images generated at different inversion times $t$ with forward-backward diffusion.** The first column represents the starting images $x_0$, while the other columns represent the generated ones $\hat{x}_0(t)$. The grid indicates the tokens represented inside the CLIP vision encoder. For inversion time $t > 0$, the red patches indicate the token embeddings that have a variation magnitude larger than a fixed threshold. These patches of variation appear in domains of growing size around the class transition, observed for $t^* \approx 0.6 \div 0.7T$ (Fig. 6).

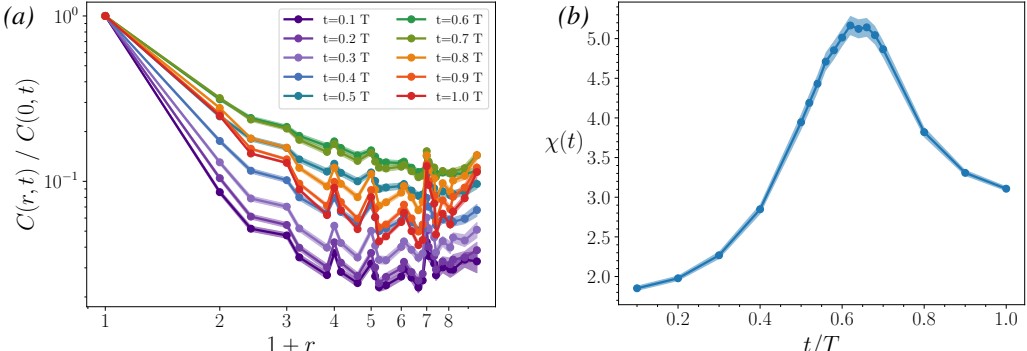

Figure 6: **Correlation measures on the variation of CLIP embeddings of images generated with forward-backward diffusion.** *(a)* The average correlation function displays a system spanning power-law behavior for $t^* \approx 0.6 \div 0.7\,T$, corresponding to the class phase transition (cf. Fig. 13). *(b)* In correspondence with the phase transition, the average susceptibility displays a peak. Data obtained with 344 starting images and 128 diffusion trajectories per starting image. The shaded areas correspond to the standard deviations over the starting images.

models. This finding highlights the compositional semantic structure of image data, similar to the phase transitions observed in language diffusion models and the RHM.

## 5   RELATED WORK

**Phase transitions in diffusion models**   Several works have studied phenomena related to phase transitions in diffusion models. Biroli et al. (2024); Ambrogioni (2023) show the presence of different dynamical regimes in the diffusion process separated by a 'speciation' cross-over where a bimodal distribution merges into a mono-modal one. Li & Chen (2024) provide bounds for critical time windows appearing in the diffusion of mixtures of strongly log-concave densities. These works do not consider hierarchical data, and thus do not present growing dynamical susceptibility or length scale at the transition.

**Hierarchical models of images and text**   Generative models have been used to describe the structure of data in several contexts, including in linguistics and signal processing. For languages, hierarchically-structured formal grammars are often used as a model of their syntactic structure (Rozenberg & Salomaa, 1997). Likewise, pattern theory formalizes the semantic decomposition of visual scenes through a hierarchy of features (Stoyan, 1997; Jin & Geman, 2006; Siskind et al., 2007; Li et al., 2009). More recently, images have been described through a hierarchical decomposition in multi-scale wavelet coefficients (Marchand et al., 2022; Kadkhodaie et al., 2023), although the underlying structure, in this case, is not tree-like.

**Semantic vs geometrical description of images**   Several studies (Rissanen et al., 2022; Wang & Vastola, 2023) pointed out that the backward diffusion process of images acts on coarse-to-fine scales. Since the Fourier spectra of images decay as power laws, higher frequencies are affected at short diffusion times, while low-frequency modes persist for longer. This is precisely the pattern we describe in the Gaussian random field model in Section 3.3 and App. B. While this viewpoint is an appropriate starting point to describe the geometrical structure at the pixel level, our hierarchical model seeks to capture a high-level, semantic description of images that we test using a CLIP encoder. This means that high/low-level features can correspond to parts of objects – such as the eyes, mouth, and nose of a face – rather than simple geometric or frequency components. The two descriptions are, therefore, complementary.

**Hierarchical models in machine learning theory**   Several studies (Mossel, 2016; Shalev-Shwartz et al., 2017; Malach & Shalev-Shwartz, 2018; 2020) have shown that the correlations between input and task are crucial for determining learnability in hierarchical data. Furthermore, the internal representations of trained deep networks reflect the data's latent structure (Cagnetta et al., 2024; Allen-Zhu & Li, 2023). Sclocchi et al. (2025) showed that diffusion on hierarchical data can be solved using Belief Propagation. Mei (2024) showed that U-Nets can efficiently approximate the Belief Propagation algorithm on hierarchical data, while Garnier-Brun et al. (2024) provided evidence that transformers can implement the same algorithm.

## 6   DISCUSSION AND CONCLUSION

In this work, we showed that when data exhibit a hierarchical structure, the changes induced by forward-backward experiments in diffusion models reveal a growing correlation length and susceptibility near a phase transition. At this critical point, changes in the data become highly correlated, reflecting changes in deep latent variables. In particular, we focused on understanding how modifications in the latent variables manifest in the data, in contrast with common approaches which attempt to reconstruct the latent representations from visible data.

Our predictions for a hierarchical model were confirmed through experiments across different natural data modalities, showing a remarkable level of universality. This supports the hypothesis that hierarchical and compositional structures are fundamental, universal properties underlying natural data as diverse as images and text.

Such fundamental analyses have the potential to impact practical applications. For example, they can enhance the interpretability of deep networks, whose representations are believed to reflect the hierarchical structure of data. Moreover, the diffusion dynamics of high and low-level features can suggest improved training strategies, for instance, to avoid mode collapse when fine-tuning diffusion models (Barceló et al., 2024).

Future work may include interpreting the large, correlated changes in text in terms of grammatical structure and context variables, possibly sharpening these concepts through the data-driven method introduced in this study. Moreover, better capturing the grammatical structure of real languages may require considering more general latent models involving context dependencies. A challenge for future work is extending our theoretical analysis to such cases.

ACKNOWLEDGMENTS

We thank Umberto Maria Tomasini for discussions at the early stages of this project. We thank Francesco Cagnetta, Daniel Korchinski, and Jack Parley for providing feedback on the manuscript. NL is supported by the EPFL AI4science program. This work was supported by a grant from the Simons Foundation (#454953 Matthieu Wyart).

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

## A    THE RANDOM HIERARCHY MODEL

**Data generation in the RHM**    A datum of the RHM is generated with the following procedure:

- a symbol at the root $h_1^{(L)}$ is chosen uniformly at random from $\mathbb{V}^{(L)}$;
- one of the $m$ production rules associated with the symbol sampled at the root is chosen uniformly at random. This choice defines the values, belonging to $\mathbb{V}^{(L-1)}$, of the $s$ children nodes $h_1^{(L-1)}, \ldots, h_s^{(L-1)}$;
- the procedure is iterated sampling the symbols at layer $\ell - 1$ according to the symbols at layer $\ell$ and their respective production rules;
- as a result, a string of $s^L$ symbols belonging to $\mathbb{V}^{(0)}$ is generated at the leaf nodes. This string is the generated datum $\boldsymbol{x}$.

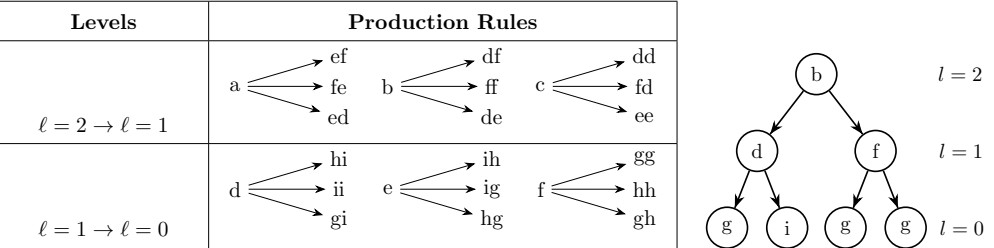

Figure 7: **Example of RHM with** $L = 2$, $s = 2$, $v = 3$, $m = 3$. *Left*: example of production rules with vocabularies $\mathbb{V}^{(2)} = \{a, b, c\}$, $\mathbb{V}^{(1)} = \{d, e, f\}$, $\mathbb{V}^{(0)} = \{g, h, i\}$. *Right*: one possible datum generated by the production rules, with the hierarchical levels indicated on the right.

### A.1    DENOISING THE RHM WITH BELIEF PROPAGATION

In the RHM, knowing the production rules of its tree structure, the Bayes-optimal denoising of its data can be done exactly using the Belief Propagation (BP) algorithm (Sclocchi et al., 2025).

In the factor graph of the RHM tree, all latent and visible variables $h_i^{(\ell)}$ represent variable nodes, while the RHM rules connecting them are factor nodes. Each variable node is associated to two BP messages, one coming from below, $\nu_\uparrow(h_i^{(\ell)})$, and one coming from above, $\nu_\downarrow(h_i^{(\ell)})$. The starting point of the BP algorithm is the definition of the messages at the boundaries of the tree, which are the upward messages at the leaves $\nu_\uparrow(h_i^{(0)})$ and the downward message at the root node $\nu_\downarrow(h_1^{(L)})$. Since we consider class-unconditional diffusion processes, we consider the latter as being uniform over the values of $\mathbb{V}^{(L)}$. The initialization at the leaves, instead, corresponds to the prior belief on the values of the single visible tokens, which is given by the noisy observation. In the case of diffusion processes, the noisy observation $\boldsymbol{x}_t$ gives prior beliefs $\nu_\uparrow(h_i^{(0)}) = p(\hat{x}_{0,i}|x_{t,i})$, which can be computed for the single token by Bayes' rule and depends on the specific diffusion process under consideration.

### A.1.1    BP ITERATION

The initialization of BP is given by the leaf messages $\nu_\uparrow(h_i^{(0)})$, $i \in [s^L]$. For each $s$-patch at level $\ell$, e.g., $\{h_i^{(\ell)}\}_{i=1,\ldots,s}$, having a common parent node at layer $\ell + 1$, e.g. $h_1^{(\ell+1)}$, the upward message in the upper level is computed as:

$$\tilde{\nu}_\uparrow\left(h_1^{(\ell+1)} = y\right) = \sum_{a_1,\ldots,a_s \in \mathbb{V}^{(\ell)\otimes s}} \psi^{(\ell+1)}(y, a_1, \ldots, a_s) \prod_{i=1}^s \nu_\uparrow\left(h_i^{(\ell)} = a_i\right), \qquad (8)$$

$$\nu_\uparrow(h_1^{(\ell+1)} = y) = \frac{\tilde{\nu}_\uparrow(h_1^{(\ell+1)} = y)}{\sum_{y' \in \mathbb{V}^{(\ell+1)}} \tilde{\nu}_\uparrow(h_1^{(\ell+1)} = y')}, \qquad (9)$$

where the factor $\psi^{(\ell+1)}(y, a_1, \ldots, a_s)$ reads

$$\psi^{(\ell+1)}(y, a_1, ..., a_s) = \begin{cases} 1, & \text{if } y \to (a_1, ..., a_s) \text{ is a rule at layer } (\ell+1) \to \ell \\ 0, & \text{otherwise.} \end{cases} \tag{10}$$

This upward process is iterated from the leaf nodes at $\ell = 0$ until the root node at $\ell = L$. Afterward, BP computes the downward messages. The initialization at the root node is given by a uniform prior over the symbols of $\mathbb{V}^{(L)}$, i.e.

$$\nu_\downarrow \left( \mathrm{h}_1^{(L)} = a \right) = \frac{1}{v}, \quad \forall a \in \mathbb{V}^{(L)}. \tag{11}$$

For the same $s$-patch at layer $\ell$ and parent node at layer $\ell+1$ as before, the downward message for $\mathrm{h}_1^{(\ell)}$ is given by

$$\tilde{\nu}_\downarrow(\mathrm{h}_1^{(\ell)} = a_1) = \sum_{\substack{a_2,...,a_s \in \mathbb{V}^{(\ell) \otimes (s-1)} \\ y \in \mathbb{V}^{(\ell+1)}}} \psi^{(\ell+1)}(y, a_1, ..., a_s)\, \nu_\downarrow(\mathrm{h}_1^{(\ell+1)} = y) \prod_{i=2}^{s} \nu_\uparrow(\mathrm{h}_i^{(\ell)} = a_i), \tag{12}$$

$$\nu_\downarrow(\mathrm{h}_1^{(\ell)} = a) = \frac{\tilde{\nu}_\downarrow(\mathrm{h}_1^{(\ell)} = a)}{\sum_{a' \in \mathbb{V}^{(\ell)}} \tilde{\nu}_\downarrow(\mathrm{h}_1^{(\ell)} = a')}, \tag{13}$$

with the same factor node of Eq. (10).

At the end of the upward-downward iteration, each variable node $\mathrm{h}_i^{(\ell)}$ is associated with two BP messages for each symbol of the vocabulary $\mathbb{V}^{(\ell)}$: $\nu_\uparrow(\mathrm{h}_i^{(\ell)})$ and $\nu_\downarrow(\mathrm{h}_i^{(\ell)})$. Their product gives the marginal probability of the value of the node:

$$p(\mathrm{h}_i^{(\ell)} = a) \propto \nu_\uparrow(\mathrm{h}_i^{(\ell)} = a)\, \nu_\downarrow(\mathrm{h}_i^{(\ell)} = a), \quad a \in \mathbb{V}^{(\ell)}. \tag{14}$$

These marginal probabilities are conditioned on the BP messages at the leaf nodes, which can come from a noisy observation of an RHM datum, as is the case for denoising diffusion.

Similarly, sampling from the posterior probabilities given by BP is done by starting sampling from the marginal probability at the root and then iteratively updating the marginal probabilities every time a new node is sampled (Mezard & Montanari, 2009).

### A.1.2 Priors at the Leaves

**Masking diffusion** Let's consider a datum $\boldsymbol{x}_0$ of the RHM undergoing masking diffusion. At any time $t$, the tokens of $\boldsymbol{x}_t$ can have value

$$\begin{aligned} x_{t,i} &= x_{0,i}, & \text{if token } i \text{ has not yet been masked;} \\ x_{t,i} &= [\mathsf{MASK}], & \text{if token } i \text{ has already been masked.} \end{aligned} \tag{15}$$

Therefore, given the noisy observation $\boldsymbol{x}_t$, the prior belief $\nu_\uparrow(\mathrm{h}_i^{(0)} = a)$ on the value of the token $i$ being equal to $a$ is given by $p(x_{0,i}|x_{t,i})$, that is:

$$\begin{aligned} \nu_\uparrow \left( \mathrm{h}_i^{(0)} = a \right) &= \delta_{a,\bar{a}}, & \text{if } x_{t,i} = \bar{a} \in \mathbb{V}^{(0)}; \\ \nu_\uparrow \left( \mathrm{h}_i^{(0)} = a \right) &= 1/v, & \forall a \in \mathbb{V}^{(0)} \text{ if } x_{t,i} = [\mathsf{MASK}]. \end{aligned} \tag{16}$$

$\epsilon$**-process** In this process, instead of running a forward diffusion process, we act directly on the leaf priors. We introduce a noise-to-signal ratio $\epsilon \in [0, 1]$, which controls the noise level instead of the diffusion time $t$. Starting from a datum $\boldsymbol{x}_0$, whose $i$-th token has value $\bar{a} \in \mathbb{V}^{(0)}$, the prior beliefs at the leaf node $i$ taking values in $\mathbb{V}^{(0)}$ are defined as

$$\begin{cases} \nu_\uparrow \left( \mathrm{h}_i^{(0)} = \overline{a} \right) & = 1 - \epsilon + \epsilon/v, \qquad \text{for } x_{0,i} = \overline{a}; \\ \nu_\uparrow \left( \mathrm{h}_i^{(0)} = a \right) & = \epsilon/v, \qquad\qquad \forall a \in \mathbb{V}^{(0)} \setminus \overline{a}. \end{cases} \qquad (17)$$

The role of $\epsilon$ is decreasing the prior belief on the starting value of a token. This process can be interpreted as an averaged forward diffusion process, where the average is made over different forward trajectories. In the example of masking diffusion (Eq. (16)), calling $1 - \alpha_t$ the probability of a token being masked at time $t$, the average prior beliefs at the leaves read

$$\begin{cases} \left\langle \nu_\uparrow(\mathrm{h}_i^{(0)} = \overline{a}) \right\rangle & = \alpha_t + \frac{1-\alpha_t}{v}, \qquad \text{where } x_{0,i} = \overline{a}; \\ \left\langle \nu_\uparrow(\mathrm{h}_i^{(0)} = a) \right\rangle & = \frac{1-\alpha_t}{v}, \qquad\qquad \forall a \in \mathbb{V}^{(0)} \setminus \overline{a}, \end{cases} \qquad (18)$$

which have the same functional form as Eq. (17) by identifying $\epsilon = 1 - \alpha_t$. Both $\epsilon$ and $1 - \alpha_t$ vary between $0$ and $1$ and play the role of noise-to-signal ratio in their respective processes. However, the fluctuations of the upward beliefs around their mean in the masking diffusion change the statistics of the BP messages propagating upwards and make the mapping $\epsilon = 1 - \alpha_t$ inaccurate. For example, in the experimental data of Fig. 2, the phase transition in the $\epsilon$-process is located at $\epsilon^* \simeq 0.74$, while it is found at $1 - \alpha_{t^*} = t^*/T \simeq 0.3$ for masking diffusion.

### A.1.3   BP SAMPLING VS BACKWARD DIFFUSION

**BP sampling**   As discussed at the end of section A.1.1, BP allows for sampling directly from the posterior probability $p(\hat{x}_0|x_t)$. Given a noisy observation $x_t$ and the corresponding marginal probabilities $p(\mathrm{h}_i^{(\ell)}|x_t)$, the sampling procedes as follows:

- a root symbol $\mathrm{h}_1^{(L)} = \hat{y}$, $\hat{y} \in \mathbb{V}^{(L)}$, is sampled according to the probability $p(\mathrm{h}_1^{(L)}|x_t)$;
- the corresponding downward message is updated as $\nu_\downarrow \left( \mathrm{h}_1^{(L)} = y \right) = \delta_{y,\hat{y}}$;
- the probabilities of the production rules $y \to (a_1, ..., a_s)$ form layer $L$ to layer $L - 1$ are computed as

$$p \left( y \to a_1, ..., a_s | x_t, \mathrm{h}_1^{(L)} = \hat{y} \right) \propto \nu_\downarrow \left( \mathrm{h}_1^{(L)} = y \right) \nu_\uparrow \left( \mathrm{h}_1^{(L-1)} = a_1 \right) \cdot ... \cdot \nu_\uparrow \left( \mathrm{h}_s^{(L-1)} = a_s \right) . \qquad (19)$$

  Notice that the upward messages $\nu_\uparrow \left( \mathrm{h}_i^{(L-1)} = a_i \right)$ carry the information on the observation $x_t$;
- a production rule $y \to (a_1, ..., a_s)$ is sampled according to the probabilities of Eq. (19). This gives the values $\hat{a}_i \in \mathbb{V}^{(L-1)}$ of the latent nodes $\mathrm{h}_i^{(L-1)}$. The corresponding downward messages are updated as $\nu_\downarrow \left( \mathrm{h}_i^{(L-1)} = a \right) = \delta_{a,\hat{a}_i}$;
- the probabilities of the production rules from layer $L - 1$ to $L - 2$ are computed as in Eq. (19);
- the sampling procedure continuous up to the visible layer $\mathrm{h}_i^{(0)}$, giving a leaf configuration $\hat{x}_0$.

The obtained sequence $\hat{x}_0$ is a configuration of the RHM sampled from the posterior $p(\hat{x}_0|x_t)$.

**Backward diffusion with BP**   The BP sampling above is equivalent to running the backward dynamics with the true score function of the RHM. In fact, given a noisy observation $x_t$ at time $t$, the marginal probabilities $p(\mathrm{h}_i^{(\ell)} = a|x_t)$ at the visible nodes can be used to compute the expectation values $\mathbb{E}(\mathrm{h}_i^{(\ell)}|x_t)$, which corresponds to $\mathbb{E}(\hat{x}_0|x_t)$. This expectation gives the score function at $x_t$ at time $t$, which can be used in the backward dynamics to sample $x_{t-1}$ at time $t - 1$, and so on.

Fig. 8 compares BP sampling and the backward diffusion with the exact score function in the case of masking diffusion. Both the average correlation functions and the dynamical susceptibility at different masking fractions $t/T$ show the same behavior, independently of the sampling procedure.

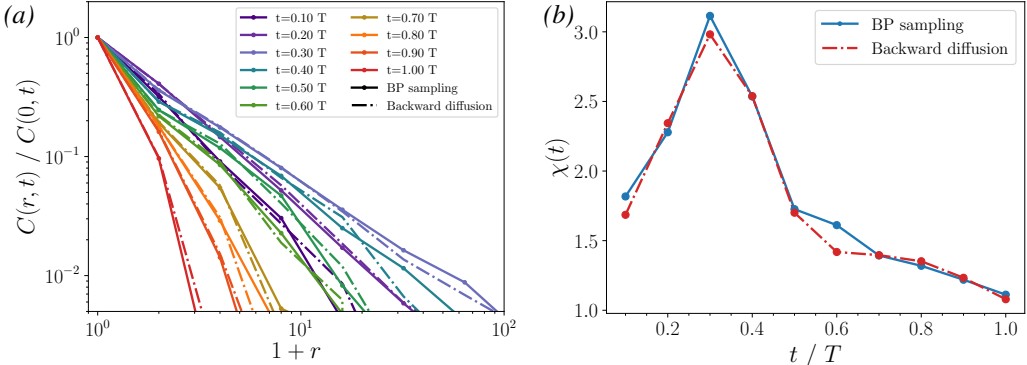

Figure 8: **Comparison between BP sampling and backward diffusion for masking in the Random Hierarchy Model (RHM).** Forward-backward experiments with masking diffusion, where the sampling from the posterior $p(\hat{x}_0|x_t)$ is done with BP sampling (continuous lines) or by running the backward diffusion dynamics (dotted-dashed lines), using the score function given by BP. Both the average correlation functions of changes (panel *(a)*) and the dynamical susceptibility (panel *(b)*) for different masking fractions $t/T$ do not depend on the sampling process. Data for RHM parameters $v = 32$, $m = 8$, $s = 2$, $L = 8$, averaged over 32 starting data and 256 diffusion trajectories per starting datum.

## A.2 MEAN-FIELD THEORY OF THE $\epsilon$-PROCESS

**Computation of the marginal probabilities** Starting from Eq. (17) and the BP iterative Eqs. (8), (9), (12) and (13), Sclocchi et al. (2025) computed the average BP messages at each layer $\ell$, where the average is performed over the possible choices of the RHM rules. The result consists in the average messages associated with reconstructing the starting value $\overline{a} \in \mathbb{V}^{(\ell)}$ of a latent node $h_i^{(\ell)}$,

$$\left\langle \nu_\uparrow \left( h_i^{(\ell)} = \overline{a} \right) \right\rangle_\psi = p_\ell, \quad \left\langle \nu_\downarrow \left( h_i^{(\ell)} = \overline{a} \right) \right\rangle_\psi = q_\ell, \tag{20}$$

where the average $\langle \dots \rangle_\psi$ is performed over the factor nodes $\psi$ representing the randomly chosen rules of the RHM. The values of $p_\ell$ and $q_\ell$ can be computed layer-by-layer through the following iterative maps:

$$p_{\ell+1} = F(p_\ell), \quad q_{\ell-1} = G(q_\ell, p_{\ell-1}), \tag{21}$$

where

$$F(p) = \frac{p^s + f\frac{m-1}{mv-1}(1-p^s)}{p^s + f(1-p^s)}, \tag{22}$$

$$G(q, p) = \frac{q\,p^{s-1} + f\frac{m-q}{mv-1}(1-p^{s-1})}{q\,p^{s-1} + f\frac{m-q}{mv-1}(1-p^{s-1}) + (v-1)f\frac{m-q}{mv-1}}, \tag{23}$$

and $f = \frac{mv-1}{v^s-1}$. The initial conditions are given by

$$p_0 = 1 - \epsilon + \epsilon/v, \tag{24}$$

$$q_L = 1/v. \tag{25}$$

Notice that the expectation values $p_\ell$ and $q_\ell$ only depend on the layer $\ell$ and not on the specific position of the node $i$ inside the layer. Once $p_\ell$ and $q_\ell$ have been computed for every layer $\ell = 0, \dots, L$, the average marginal probability of reconstructing the original value $\overline{a} \in \mathbb{V}^{(\ell)}$ of the variable $h^{(\ell)}$ is given by

$$P(h^{(\ell)} = \overline{a}) = \frac{p_\ell q_\ell}{p_\ell q_\ell + \frac{(1-p_\ell)(1-q_\ell)}{v-1}}. \tag{26}$$

This marginal probability is conditioned on the initialization of the leaf nodes (Eq. (17)) and only depends on the layer $\ell$, not on the position of the node inside the layer. Given the initialization of $q_L$, the probability of reconstructing the root node $P(\mathrm{h}^{(L)} = \overline{a})$, that is the class of the datum, is given by

$$P(\mathrm{h}^{(L)} = \overline{a}) = p_L. \tag{27}$$

Therefore, in the limit of large depth $L \to \infty$, the value of $p_L$ is given by one of the fixed of the iterative map $F(p)$. When $F'(1) > 1$, $F(p)$ has two fixed points: $p = 1$, which is repulsive, and $p = 1/v$, which is attractive. This implies that, in this regime, for any noise level $\epsilon > 0$ at the leaf nodes, it is impossible to reconstruct the value of the class better than random chance. Instead, when $F'(1) < 1$, that is

$$s\, m\, \frac{v-1}{v^s - 1} < 1, \tag{28}$$

a third non-trivial fixed point $p^* = F(p^*)$ appears, which is repulsive, while both $p = 1$ and $p = 1/v$ are now attractive. This implies the presence of a phase transition at a specific noise level $\epsilon^* = \frac{1 - p^*}{1 - 1/v}$. For $\epsilon < \epsilon^*$, the class is reconstructed, for $\epsilon > \epsilon^*$ it is not.

**Computation of the correlation functions** Similar to the marginal probabilities, the average correlation function can also be computed through an annealed average over the RHM rules. Let's consider two leaf nodes $\mathrm{h}_i^{(0)}$ and $\mathrm{h}_j^{(0)}$ connected to the common ancestor $\mathrm{h}_1^{(\tilde{\ell})}$ at layer $\tilde{\ell}$ through the nodes $\mathrm{h}_1^{(\tilde{\ell}-1)}$ and $\mathrm{h}_2^{(\tilde{\ell}-1)}$. Given the tree structure, their joint probability distribution can be written as

$$P(\mathrm{h}_i^{(0)}, \mathrm{h}_j^{(0)}) =$$
$$\sum_{\mathrm{h}_l^{(\tilde{\ell}-1)}, \mathrm{h}_m^{(\tilde{\ell}-1)}} P\left(\mathrm{h}_i^{(0)} | \mathrm{h}_l^{(\tilde{\ell}-1)}\right) P\left(\mathrm{h}_j^{(0)} | \mathrm{h}_m^{(\tilde{\ell}-1)}\right) \sum_{\mathrm{h}_k^{(\tilde{\ell})}} P\left(\mathrm{h}_l^{(\tilde{\ell}-1)}, \mathrm{h}_m^{(\tilde{\ell}-1)} | \mathrm{h}_k^{(\tilde{\ell})}\right) P\left(\mathrm{h}_k^{(\tilde{\ell})}\right) \tag{29}$$

In the mean-field approach, the average joint probability only depends on the tree-distance $\tilde{\ell}$ between $i$ and $j$ and not their precise location. Moreover, we are only interested in the probability that both the starting values of $\mathrm{h}_i^{(0)}$, $\mathrm{h}_j^{(0)}$ are reconstructed, and the probability of only one of the two is reconstructed. In the following, we use an overline $\overline{\cdots}$ to indicate the starting value of a variable to be reconstructed. We need to compute

$$\left\langle P(\mathrm{h}_i^{(0)} = \overline{a}_i, \mathrm{h}_j^{(0)} = \overline{a}_j) \right\rangle_\psi, \tag{30}$$

$$\left\langle P(\mathrm{h}_i^{(0)} = \overline{a}_i, \mathrm{h}_j^{(0)} \neq \overline{a}_j) \right\rangle_\psi = \left\langle P(\mathrm{h}_i^{(0)} \neq \overline{a}_i, \mathrm{h}_j^{(0)} = \overline{a}_j) \right\rangle_\psi, \tag{31}$$

where the average $\langle \ldots \rangle_\psi$ is done over the possible choices of RHM rules. Using the same strategy for the computation of the marginal probabilities, we compute the average of each term in Eq. (29) by substituting the BP messages with their averages. For this purpose, we first define the average marginal conditioned on the downward messages at layer $\hat{\ell}$, $P(\mathrm{h}^{(\ell)} = \overline{a}^\ell | q_{\hat{\ell}} = c)$, with $\ell < \hat{\ell}$. This is computed with Eq. (26) by iterating the equations 21 between layers 0 and $\hat{\ell}$ and using the initial conditions of Eq. (24) and $q_{\hat{\ell}} = c$. Therefore, the marginals of Eq. (26) correspond to $P(\mathrm{h}^{(\ell)} = \overline{a}^\ell | q_L = 1/v)$. For the marginals in Eq. (29) we have:

$$\langle P\left(\mathrm{h}_k^{(\tilde{\ell})} = \overline{a}_k^{(\tilde{\ell})}\right)\rangle_\psi = P(\mathrm{h}^{(\tilde{\ell})} = \overline{a}^{(\tilde{\ell})} | q_L = 1/v), \tag{32}$$

that is the average marginal computed in Eq. (26);

$$\left\langle P\left(\mathrm{h}_i^{(0)} = \overline{a}_i | \mathrm{h}_l^{(\tilde{\ell}-1)} = \overline{a}_l^{(\tilde{\ell}-1)}\right) \right\rangle_\psi = P(\mathrm{h}^{(0)} = \overline{a} | q_{\tilde{\ell}-1} = 1), \tag{33}$$

$$\left\langle P\left(\mathrm{h}_i^{(0)} = \overline{a}_i | \mathrm{h}_l^{(\tilde{\ell}-1)} \neq \overline{a}_l^{(\tilde{\ell}-1)}\right) \right\rangle_\psi = P(\mathrm{h}^{(0)} = \overline{a} | q_{\tilde{\ell}-1} = 0). \tag{34}$$

The probability terms of the type $P(\mathrm{h}^{(0)} \neq \overline{a} | \dots)$ are given by $1 - P(\mathrm{h}^{(0)} \neq \overline{a} | \dots)$. Since these averages only depend on the layer level $\tilde{\ell}$, they are the same for $\left\langle P\left(\mathrm{h}_j^{(0)} | \mathrm{h}_m^{(\tilde{\ell}-1)}\right)\right\rangle_{\psi}$. The last term to compute is the joint $P\left(\mathrm{h}_l^{(\tilde{\ell}-1)}, \mathrm{h}_m^{(\tilde{\ell}-1)} | \mathrm{h}_k^{(\tilde{\ell})}\right)$ which can be expressed in terms of BP messages:

$$P\left(\mathrm{h}_l^{(\tilde{\ell}-1)} = a_l, \mathrm{h}_m^{(\tilde{\ell}-1)} = a_m | \mathrm{h}_k^{(\tilde{\ell})} = y\right) \propto$$

$$\sum_{a_{m+1}, \dots, a_s \in \mathbb{V}^{(\ell) \otimes (s-2)}} \psi^{(\ell)}(y, a_l, a_m, \dots, a_s) \, \nu_\uparrow(\mathrm{h}_l^{(\ell)} = a_l) \, \nu_\uparrow(\mathrm{h}_m^{(\ell)} = a_m) \prod_{i \neq l, m}^{s} \nu_\uparrow(\mathrm{h}_i^{(\ell)} = a_i). \tag{35}$$

Computing the averages over the rules, we have:

$$\left\langle P\left(\mathrm{h}_l^{(\tilde{\ell}-1)} = \overline{a}_l, \mathrm{h}_m^{(\tilde{\ell}-1)} = \overline{a}_m | \mathrm{h}_k^{(\tilde{\ell})} = \overline{y}\right)\right\rangle_{\psi} = p_{\ell-1}^2 \, / \, Z_{\overline{y}}^{(\tilde{\ell}-1)}, \tag{36}$$

$$\left\langle P\left(\mathrm{h}_l^{(\tilde{\ell}-1)} = \overline{a}_l, \mathrm{h}_m^{(\tilde{\ell}-1)} \neq \overline{a}_m | \mathrm{h}_k^{(\tilde{\ell})} = \overline{y}\right)\right\rangle_{\psi} = f \, p_{\ell-1}(1 - p_{\ell-1}) \frac{m-1}{mv - 1} \, / \, Z_{\overline{y}}^{(\tilde{\ell}-1)}, \tag{37}$$

$$\left\langle P\left(\mathrm{h}_l^{(\tilde{\ell}-1)} \neq \overline{a}_l, \mathrm{h}_m^{(\tilde{\ell}-1)} \neq \overline{a}_m | \mathrm{h}_k^{(\tilde{\ell})} = \overline{y}\right)\right\rangle_{\psi} = f \, (1 - p_{\ell-1})^2 \frac{m-1}{mv - 1} \, / \, Z_{\overline{y}}^{(\tilde{\ell}-1)}, \tag{38}$$

$$Z_{\overline{y}}^{(\tilde{\ell}-1)} = p_{\ell-1}^2 + f \frac{m-1}{mv - 1}(1 - p_{\ell-1}^2) \tag{39}$$

$$\left\langle P\left(\mathrm{h}_l^{(\tilde{\ell}-1)} = \overline{a}_l, \mathrm{h}_m^{(\tilde{\ell}-1)} = \overline{a}_m | \mathrm{h}_k^{(\tilde{\ell})} \neq \overline{y}\right)\right\rangle_{\psi} = 0, \tag{40}$$

$$\left\langle P\left(\mathrm{h}_l^{(\tilde{\ell}-1)} = \overline{a}_l, \mathrm{h}_m^{(\tilde{\ell}-1)} \neq \overline{a}_m | \mathrm{h}_k^{(\tilde{\ell})} \neq \overline{y}\right)\right\rangle_{\psi} = f \, p_{\ell-1}(1 - p_{\ell-1}) \frac{m}{mv - 1} \, / \, Z_y^{(\tilde{\ell}-1)}, \tag{41}$$

$$\left\langle P\left(\mathrm{h}_l^{(\tilde{\ell}-1)} \neq \overline{a}_l, \mathrm{h}_m^{(\tilde{\ell}-1)} \neq \overline{a}_m | \mathrm{h}_k^{(\tilde{\ell})} \neq \overline{y}\right)\right\rangle_{\psi} = f \, (1 - p_{\ell-1})^2 \frac{m}{mv - 1} \, / \, Z_y^{(\tilde{\ell}-1)}, \tag{42}$$

$$Z_y^{(\tilde{\ell}-1)} = f \frac{m}{mv - 1}(1 - p_{\ell-1}^2) \tag{43}$$

We can combine these terms with the marginals Eq. (32) to obtain $\left\langle P\left(\mathrm{h}_l^{(\tilde{\ell}-1)}, \mathrm{h}_m^{(\tilde{\ell}-1)}\right)\right\rangle_{\psi}$. We can write this probabilities in a $2 \times 2$ matrix $\boldsymbol{C}^{(\tilde{\ell}-1)}$ such that:

$$C_{11}^{(\tilde{\ell}-1)} = \left\langle P\left(\mathrm{h}_l^{(\tilde{\ell}-1)} = \overline{a}_l, \mathrm{h}_m^{(\tilde{\ell}-1)} = \overline{a}_m\right)\right\rangle_{\psi}, \tag{44}$$

$$C_{12}^{(\tilde{\ell}-1)} = C_{21}^{(\tilde{\ell}-1)} = \left\langle P\left(\mathrm{h}_l^{(\tilde{\ell}-1)} = \overline{a}_l, \mathrm{h}_m^{(\tilde{\ell}-1)} \neq \overline{a}_m\right)\right\rangle_{\psi}, \tag{45}$$

$$C_{22}^{(\tilde{\ell}-1)} = \left\langle P\left(\mathrm{h}_l^{(\tilde{\ell}-1)} \neq \overline{a}_l, \mathrm{h}_m^{(\tilde{\ell}-1)} \neq \overline{a}_m\right)\right\rangle_{\psi}. \tag{46}$$

Similarly, also the conditional marginals of Eqs. (33) and (34) can be written as a $2 \times 2$ matrix $\boldsymbol{T}^{(\tilde{\ell}-1)}$:

$$T_{11}^{(\tilde{\ell}-1)} = \left\langle P\left(\mathrm{h}_i^{(0)} = \overline{a}_i | \mathrm{h}_l^{(\tilde{\ell}-1)} = \overline{a}_l^{(\tilde{\ell}-1)}\right)\right\rangle_{\psi}, \tag{47}$$

$$T_{12}^{(\tilde{\ell}-1)} = \left\langle P\left(\mathrm{h}_i^{(0)} = \overline{a}_i | \mathrm{h}_l^{(\tilde{\ell}-1)} \neq \overline{a}_l^{(\tilde{\ell}-1)}\right)\right\rangle_{\psi}, \tag{48}$$

$$T_{21}^{(\tilde{\ell}-1)} = \left\langle P\left(\mathrm{h}_i^{(0)} \neq \overline{a}_i | \mathrm{h}_l^{(\tilde{\ell}-1)} = \overline{a}_l^{(\tilde{\ell}-1)}\right)\right\rangle_{\psi}, \tag{49}$$

$$T_{22}^{(\tilde{\ell}-1)} = \left\langle P\left(\mathrm{h}_i^{(0)} \neq \overline{a}_i | \mathrm{h}_l^{(\tilde{\ell}-1)} \neq \overline{a}_l^{(\tilde{\ell}-1)}\right)\right\rangle_{\psi}. \tag{50}$$

Collecting the values of $\langle P(\mathrm{h}_i^{(0)}, \mathrm{h}_j^{(0)})\rangle_\psi$ into a $2 \times 2$ matrix $\boldsymbol{P}(\mathrm{h}_i^{(0)}, \mathrm{h}_j^{(0)})$, we finally obtain

$$\boldsymbol{P}(\mathrm{h}_i^{(0)}, \mathrm{h}_j^{(0)}) = \boldsymbol{T}^{(\tilde{\ell}-1)} \, \boldsymbol{C}^{(\tilde{\ell}-1)} \, \boldsymbol{T}^{(\tilde{\ell}-1)\top}. \tag{51}$$

In the language of the spin variables introduced in Section 3, the probability of reconstructing a variable $\mathrm{h}_i^{(0)} = \bar{a}_i$ is the probability that $\sigma_i^0 = +1$, while $\mathrm{h}_i^{(0)} \neq \bar{a}_i$ corresponds to $\sigma_i^0 = -1$.

### A.2.1 Dynamical correlation length

In the mean-field approach, the average upward belief $p_\ell$ in the original value of a latent variable at layer $\ell$ can be computed through the iterative map

$$p_\ell = F(p_{\ell-1}), \tag{52}$$

where the functional form of $F(p)$ is reported in Eq. (22) and the initial condition $p_0$ depends on the noise level $\epsilon$ as $p_0 = 1 - \epsilon + \epsilon/v$. In the limit of large depth $L \to \infty$, the probability $p_L$ of reconstructing the class is given by the fixed points of $F(p)$. For RHM parameters such that Eq. (28) is satisfied, $p_L$ undergoes a phase transition and $F(p)$ has three fixed points: two attractive ones, corresponding to $p = 1/v$ and $p = 1$, and a repulsive one, corresponding to the non-trivial solution of $p^* = F(p^*)$ with $p^* \in (\frac{1}{v}, 1)$. $p^*$ corresponds to a critical noise level $\epsilon^* = \frac{1-p^*}{1-1/v}$.

In the vicinity of $\epsilon^*$ and the limit $L \to \infty$, we can estimate the typical distance over which token changes are correlated by computing the number of layers $\tilde{\ell}$ after which the upward probability of reconstructing the latent variables $p_{\tilde{\ell}}$ approaches one of the two trivial fixed points $p = 1$ and $p = 1/v$. This corresponds to the number of layers required to escape the repulsive fixed point $p^*$.

Given the iterative map of Eq. (52), we can linearize it around the fixed point $p^*$ and iterate for $\ell$ layers,

$$\Delta p_\ell = \left(\left.\frac{dF(p)}{dp}\right|_{p^*}\right)^\ell \Delta p_0, \tag{53}$$

where $\Delta p_\ell = p_\ell - p^*$. We have that $\left.\frac{dF(p)}{dp}\right|_{p^*} > 1$ and we use the shorthand notation $F_*' = \left.\frac{dF(p)}{dp}\right|_{p^*}$. We want to compute the depth $\tilde{\ell}$ at which $F_*'^{\tilde{\ell}} |\Delta p_0| = \mathcal{O}(1)$. In terms of the corruption noise $\epsilon$, we have $F_*'^{\tilde{\ell}} |\Delta \epsilon| = \mathcal{O}(1)$, where $\Delta \epsilon = \epsilon - \epsilon^*$. Hence, $\tilde{\ell} \sim -\log|\epsilon - \epsilon^*|/\log F_*'$. From the depth $\tilde{\ell}$, we can compute the correlation length in input space as

$$\xi \simeq s^{\tilde{\ell}} \sim |\epsilon - \epsilon^*|^{-\nu} \quad \text{with } \nu = \frac{\log s}{\log F_*'}, \tag{54}$$

that diverges at the critical point: $\lim_{\epsilon \to \epsilon^*} \xi = +\infty$.

## B  Gaussian Random Field Model

Consider $u \in [-1,1]^d$. Let $\boldsymbol{x}(u)$ denote a centered Gaussian random field defined over this domain with translational-invariant isotropic covariance function $K(u, u')$. Specifically, the field satisfies $\mathbb{E}[\boldsymbol{x}(u)] = 0$ and $\mathbb{E}[\boldsymbol{x}(u)\boldsymbol{x}(u')] = K(u, u') = c(\|u - u'\|)$, where $c$ is a function depending only on the Euclidean distance $\|u - u'\|$.

Assume that the Fourier coefficients $C(k)$ of $c(\|u - u'\|)$ satisfy, for large $\|k\|$, $C(k) = \gamma\|k\|^{-a} + o(\|k\|^{-a})$, $\|k\| \to \infty$, with $0 < a < d$. This implies that the Fourier coefficients $\mathbf{X}(k)$ are independent Gaussian random variables, $\mathbf{X}(k) \sim \mathcal{N}(0, \sigma_k^2)$ with $\sigma_k^2 \asymp \|k\|^{-a}$.

### B.1  Forward-backward experiments in Fourier space

Given the independence of the Fourier coefficients $\mathbf{X}(k)$, we apply the diffusion dynamics to each Fourier coefficient independently. The noising process is given by:

$$\mathbf{X}(k)_t = \sqrt{1 - \beta_t}\mathbf{X}(k)_{t-1} + \sqrt{\beta_t}\eta, \quad \eta \sim \mathcal{N}(0, 1), \tag{55}$$

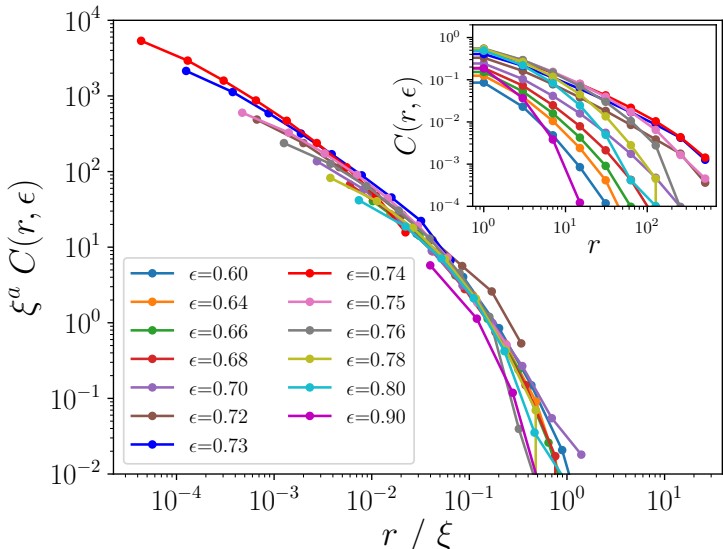

Figure 9: $\epsilon$**-process in the RHM** ($v = 32$, $m = 8$, $s = 2$, $L = 9$)**: correlation function with respect to the token distance $r$, for noise levels $\epsilon$ close to the transition $\epsilon^* \simeq 0.74$.** *(Inset)* The correlation function displays system-spanning power-law decay at the transition $\epsilon^* \simeq 0.74$, while it decays faster for noise values $\epsilon \neq \epsilon^*$. The length scale at which it departs from the critical behaviour defines the correlation length $\xi$. *(Main)* Rescaling the distance $r$ with $\xi$ given by Eq. (54) and $C(r, \epsilon)$ with $\xi^a$, $a = 1$, the different correlation functions collapse on a single curve. This implies that the power-law scaling $\xi \sim |\Delta\epsilon|^{-\nu}$ of Eq. (54) describes well the peaking of the correlation length around the class transition. For this choice of RHM parameters, $\nu \simeq 1.78$. The exponent $a = 1$ is obtained by fitting the critical decay $C(r, \epsilon^*) \sim r^{-a}$ from the data.

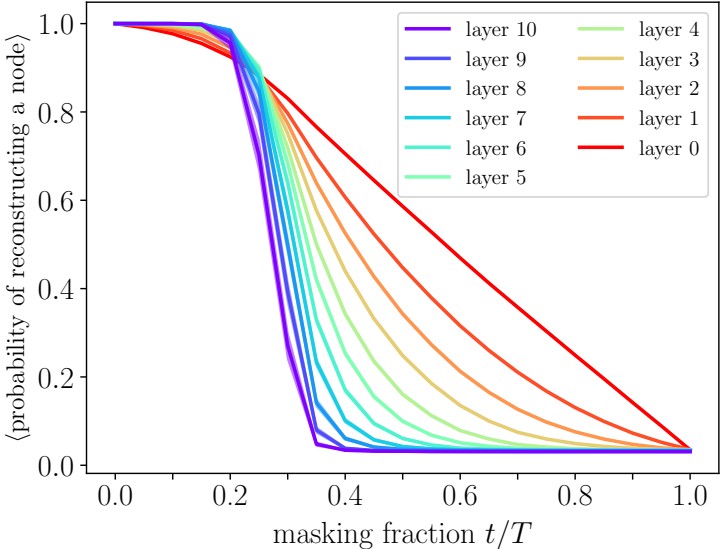

Figure 10: **Masking diffusion in the RHM: probability of reconstructing a (latent) node as a function of the inversion time $t$.** This is proportional to the masking fraction $t/T$. The probability is averaged over the nodes at a given layer. The probability of reconstructing a leaf node (layer 0) decreases smoothly with the inversion time, while the probability of reconstructing the root node (layer 10), that is the datum class, undergoes a sharper decay from 1 to $1/v$ at a critical time $t^* \simeq 0.2 \div 0.3\, T$. This sharp decay is expected to become a step-like transition in the limit of infinite depth $L \to \infty$. Data for RHM parameters $v = 32$, $m = 8$, $s = 2$, $L = 10$, averaged over 10 diffusion trajectories per 10 starting data $\boldsymbol{x}_0$.

for $t = 1, 2, \ldots, T$, where $\beta_t \in (0, 1)$ are the diffusion coefficients and $\eta$ are independent standard Gaussian variables.

By unrolling the recursion, the *forward dynamics* can be expressed as

$$\mathbf{X}(k)_t = \sqrt{\overline{\alpha_t}}\mathbf{X}(k)_0 + \sqrt{1 - \overline{\alpha_t}}\eta, \quad \eta \sim \mathcal{N}(0, 1), \tag{56}$$

where $\overline{\alpha_t} = \prod_{t'=1}^{t}(1 - \beta_{t'})$.

We then reverse the process at time $t$, following the *backward dynamics*:

$$\mathbf{X}(k)_{t-1} = \frac{1}{\sqrt{\overline{\alpha_t}}}\left(\mathbf{X}(k)_t + \beta_t \nabla_{\mathbf{X}(k)} \log q(\mathbf{X}(k)_t)\right) + \sqrt{\beta_t}z, \quad z \sim \mathcal{N}(0, 1), \tag{57}$$

where $q(\mathbf{X}(k)_t)$ is marginal probability density of $\mathbf{X}(k)_t$ in the forward process and $\nabla_{\mathbf{X}(k)} \log q(\mathbf{X}(k)_t)$ is the corresponding *score function*.

Given the forward process, $q(\mathbf{X}(k)_t)$ is Gaussian and the score function can be computed explicitly:

$$\nabla_{\mathbf{X}(k)} \log q(\mathbf{X}(k)_t) = -\frac{\mathbf{X}(k)_t}{\overline{\alpha_t}\sigma_k^2 + 1 - \overline{\alpha_t}}. \tag{58}$$

## B.2 Mode retrieval

Our goal is to determine which Fourier coefficients are retrieved after the reverse process. Specifically, we want to compute the modes $k$ for which the distance between the coefficient obtained at the end of the backward process $\widehat{\mathbf{X}}(k, t)_0 \sim p(\cdot|\mathbf{X}(k)_t)$ with the starting coefficient $\mathbf{X}(k)_0$ is small:

$$|\widehat{\mathbf{X}}(k, t)_0 - \mathbf{X}(k)_0| \ll 1. \tag{59}$$

Thus, we consider the signal-to-noise ratio (SNR) for each mode $k$

$$\mathrm{SNR}(\kappa, t) = \frac{\kappa^{-a}}{\overline{\alpha_t}^{-1} - 1}, \tag{60}$$

where $\kappa = \|k\|$.

Define the critical wavevector magnitude $\kappa^*$ where $\mathrm{SNR}(\kappa^*, t) = 1$:

$$\kappa^* = \left(\overline{\alpha_t}^{-1} - 1\right)^{-1/a} \tag{61}$$

Modes with $\kappa < \kappa^*$ (low-frequency modes) have $\mathrm{SNR} > 1$ and can be retrieved, while modes with $\kappa > \kappa^*$ (high-frequency modes) have $\mathrm{SNR} > 1$ are dominated by the noise in the forward dynamics and cannot be reconstructed.

## B.3 Correlation analysis

We seek to compute the correlation of the changes after reverting the process at time $t$. Let $\boldsymbol{x}(u, t)$ denote the field obtained after reverting the diffusion process at time $t$, at position $u$. In particular, $\boldsymbol{x}(\cdot, 0)$ denotes the starting random field. Define the difference field $\boldsymbol{z}(u, t) = \boldsymbol{x}(u, t) - \boldsymbol{x}(u, 0)$. Since the two fields are Gaussian, also $\boldsymbol{z}(\cdot, t)$ is Gaussian.

We are interested in the following spatial correlation function:

$$\mathcal{C}(r, t) = \mathbb{E}[\boldsymbol{z}(u, t)^2 \boldsymbol{z}(0, t)^2], \tag{62}$$

where $r = \|u\|$. Using Wick's theorem, we have

$$\mathcal{C}(r, t) = \mathbb{E}[\boldsymbol{z}(u, t)\boldsymbol{z}(u, t)]\,\mathbb{E}[\boldsymbol{z}(0, t)\boldsymbol{z}(0, t)] + 2\mathbb{E}[\boldsymbol{z}(u, t)\boldsymbol{z}(0, t)]^2. \tag{63}$$

The first term is a constant independent of $r$, while the second term captures the spatial dependence.

To compute $\mathbb{E}[\boldsymbol{z}(u, t)\boldsymbol{z}(0, t)]$, we express $\boldsymbol{z}(u, t)$ in terms of its Fourier coefficients $\mathbf{Z}(k, t)$. For modes with $\kappa < \kappa^*(t)$, we can assume $\mathbf{Z}(k, t) \approx 0$. For modes with $\kappa > \kappa^*(t)$, $\widehat{\mathbf{X}}(k, t)_0$ is approximately independent of $\mathbf{X}(k)_0$. Thus, $\mathbf{Z}(k, t)$ for $\kappa > \kappa^*(t)$ is a Gaussian random variable with zero mean and variance $2\sigma_k^2$.

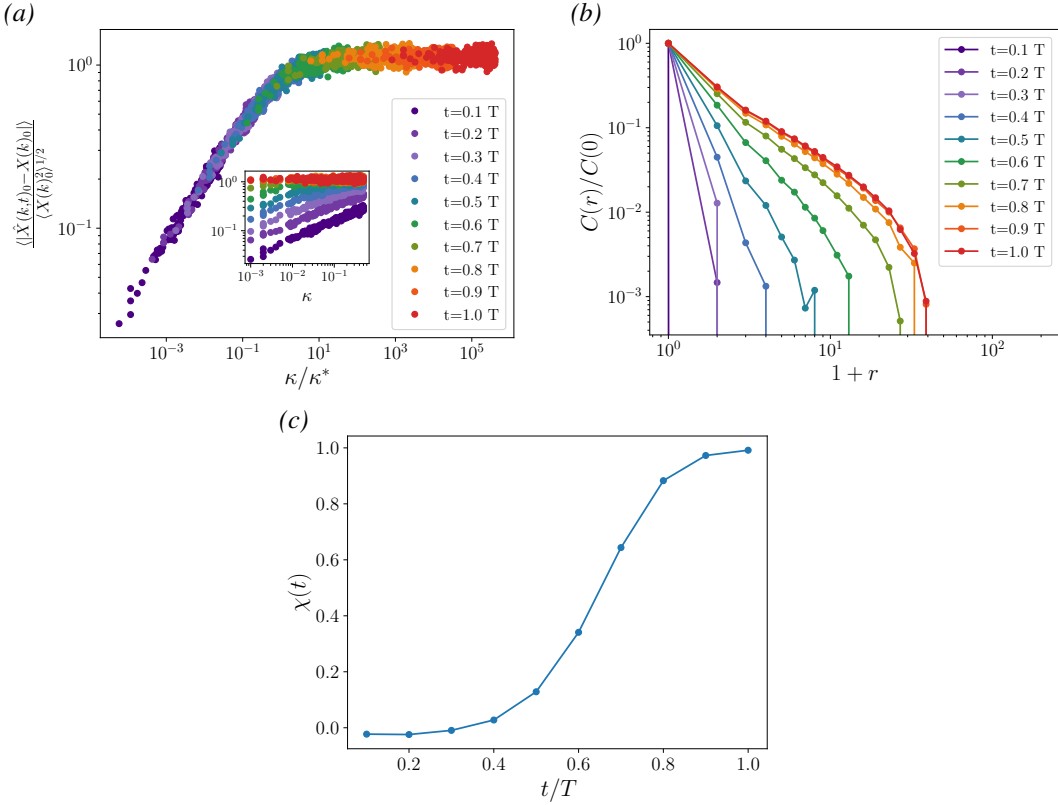

Figure 11: **Gaussian random field model.** *(a)* Relative modal errors as a function of wave-vector magnitude $|k|$. For $|k| > \kappa^*$, errors remain large, indicating unsuccessful retrieval of the Fourier coefficients, while for $|k| < \kappa^*$, the errors decrease, signifying successful recovery. *(b)* Spatial correlation function $\mathcal{C}(r, t)$, showing a power law decay at short distances and a cutoff at long distances. The correlation length increases with inversion time $t$. *(c)* The susceptibility $\chi(t)$ increases monotonically and reaches its maximum at the inversion time $t = T$.

Thus, the covariance of $\boldsymbol{z}$ is

$$\mathbb{E}[\boldsymbol{z}(u, t)\boldsymbol{z}(0, t)] \approx \int_{\|k\| > \kappa^*(t)} e^{ik^\top u} \, 2\sigma_k^2 d^d k. \tag{64}$$

Substituting $\sigma_k^2 \asymp \|k\|^{-a}$, we have:

$$\mathbb{E}[\boldsymbol{z}(u, t)\boldsymbol{z}(0, t)] \approx \int_{\|k\| > \kappa^*(t)} e^{ik^\top u} \, 2\|k\|^{-a} d^d k. \tag{65}$$

To evaluate the integral, we consider the asymptotic behavior for different regimes of $r$. At short distances $r \ll 1/\kappa^*$, the integral over $k$ is dominated by large $\kappa$ and behaves as $\mathbb{E}[\boldsymbol{z}(u, t)\boldsymbol{z}(0, t)] \approx C_1 \, r^{a-d}$, where $C_1$ is a constant. At long distances $r \gg 1/\kappa^*(t)$, the lower limit $\kappa^*(t)$ introduces an effective cutoff and the covariance decays faster than any power law.

Therefore, the correlation function $\mathcal{C}(r, t)$ exhibits algebraic decay with exponent $2(a - d)$ for $r \ll 1/\kappa^*$ and faster than any power law for $r \gg 1/\kappa^*$.

### B.4 DISCUSSION

For the Gaussian random field model, the correlation length $\xi \sim 1/\kappa^*(t)$ is a monotonically increasing function of the inversion time $t$, or noise-to-signal ratio (NSR). As a result, the susceptibility $\chi(t)$ – calculated by integrating the correlation function over space – also increases monotonically and

reaches its maximum at the inversion time $t = T$, where the NSR $= \infty$. This behavior contrasts sharply with the hierarchical data studied here, where a phase transition occurs at a finite time/NSR. As discussed in the main text, this divergence arises due to the geometry of correlations induced by the hierarchical tree structure, which is absent in the Gaussian random field model.

### B.5 NUMERICAL EXPERIMENTS

In Figure 11 (a), we plot the relative modal errors $\mathcal{E}_k = \sigma_k^{-1}|\widehat{\mathbf{X}}(k,t)_0 - \mathbf{X}(k)_0|$. For $\|k\| > \kappa^*$, the errors remain $\mathcal{O}(1)$, indicating that the coefficients are not retrieved, as predicted by our analysis. Conversely, for $\|k\| > \kappa^*$, the errors decay, indicating successful recovery of the coefficients. In panel (b), we present the correlations $\mathcal{C}(r,t)$, which exhibit a power law decay followed by a cutoff. Notably, the correlation length increases monotonically with the inversion time $t$. Finally, in panel (c), we plot the susceptibility $\chi(t)$, which reaches its maximum at $t = T$.

## C LANGUAGE DIFFUSION

### C.1 SETUP

Here, we briefly describe the particular realization of discrete diffusion used in the MDLM setting, which is detailed in (Sahoo et al., 2024).

MDLMs are a form of discrete diffusion model tailored for language generation. Unlike autoregressive (AR) models, MDLMs generate text by gradually unmasking tokens, allowing for non-sequential generation. This process is governed by a forward masking and reverse unmasking process, parameterized using a Rao-Blackwellized objective to improve performance.

**Forward Process:** The forward process is defined by progressively noising a clean input sequence $x$ using a categorical distribution:

$$q(z_t|x) = \text{Cat}(z_t; \alpha_t x + (1 - \alpha_t)m), \tag{66}$$

where $z_t$ is the latent variable at time $t$, representing the noisy version of the input sequence, $x$ is the original, clean sequence of tokens, $\text{Cat}(\cdot;\cdot)$ is a categorical distribution over the possible states, $\alpha_t$ is the noise schedule function, strictly decreasing from 1 to 0 as $t$ increases, and $m$ is a one-hot vector representing the special masked token. At each time step, a fraction of the data transitions into the masked state.

**Reverse Process and Rao-Blackwellization:** The reverse diffusion process reconstructs the original data from noisy observations. It is parameterized using a neural network approximation $x_\theta(z_t, t)$, which predicts clean tokens from noisy inputs:

$$p_\theta(z_s|z_t) = \begin{cases} \text{Cat}(z_s; z_t), & \text{if } z_t \neq m, \\ \text{Cat}\left(z_s; \frac{(1-\alpha_s)m + (\alpha_s - \alpha_t)x_\theta(z_t,t)}{1 - \alpha_t}\right), & \text{if } z_t = m. \end{cases} \tag{67}$$

where $z_s$ is the latent variable at a prior time step $s$ (with $s < t$), $x_\theta(z_t, t)$ is a neural network approximation of $x$ given the noisy observation $z_t$ at time $t$, and $p_\theta(\cdot|\cdot)$ is the model distribution approximating the true reverse process.

The training objective is a *negative evidence lower bound* (NELBO), expressed as:

$$L_{\text{diffusion}} = \sum_{i=1}^{T} \mathbb{E}_q \left[ \frac{\alpha_{t(i)} - \alpha_{s(i)}}{1 - \alpha_{t(i)}} \log\langle x_\theta(z_{t(i)}), x\rangle \right], \tag{68}$$

where $T$ is the number of diffusion steps, $\alpha_{t(i)}$, $\alpha_{s(i)}$ is the noise schedules evaluated at time steps $t(i)$ and $s(i)$, respectively, $\mathbb{E}_q$ is the expectation over the forward process defined by $q$, and $\langle x_\theta(z_{t(i)}), x\rangle$ is the dot product between the neural network output $x_\theta(z_{t(i)})$ and the original input $x$.

**Continuous-Time Likelihood Bounds:**    To achieve a tighter approximation to the ELBO, the discrete objective is extended to continuous time as:

$$L_{\infty \text{NELBO}} = \mathbb{E}_q \int_0^1 \frac{\alpha_t'}{1 - \alpha_t} \log \langle x_\theta(z_t, t), x \rangle \, dt. \tag{69}$$

where $\alpha_t'$ is the time derivative of the noise schedule $\alpha_t$. The integral evaluates the objective over continuous time, providing a tighter bound on the likelihood. This formulation is invariant to the specific functional form of the noise schedule $\alpha_t$, highlighting the robustness of the MDLM approach.

**Connection to Masked Language Models:**    MDLMs leverage a masked diffusion approach where the training objective is a weighted average of classical masked language modeling (MLM) losses:

$$L_{\infty \text{NELBO}} = \mathbb{E}_q \int_0^1 \frac{\alpha_t'}{1 - \alpha_t} \sum_\ell \log \langle x_\theta^\ell(z_t), x^\ell \rangle \, dt, \tag{70}$$

where $x^\ell$: The $\ell$-th token in the original sequence, $x_\theta^\ell(z_t)$: The neural network's prediction for the $\ell$-th token given the noisy sequence $z_t$. The summation runs over all tokens in the sequence, effectively establishing a connection between MDLMs and BERT-style encoders while equipping them with generative capabilities.

We employ the MDLM proposed in (Sahoo et al., 2024) to conduct the forward-backward experiments described in Section 4, by first drawing random texts of a fixed token length from the `WikiText2` database, masking a fixed fraction of the tokens $t$, and then performing the backward diffusion process by using the masked sequence as the initial point for the MDLM model.

## C.2    EXAMPLES OF TEXT SAMPLES FOR THE FORWARD-BACKWARD EXPERIMENTS

Below, we provide examples of texts generated by the forward-backward process using MDLM seeded from `WikiText2` examples for different masking fractions. Selected samples were shown in the main text in Fig. 4 (a). We dub the text results after the forward-backward process as *U-turn* samples. As can be seen by the color coding, correlated blocks of words change together along the denoising process, as described in Section 3, and the semantic meaning of the paragraphs themselves change along the phase transition. In blue we denote masked tokens that have changed their value after the backward process, while in green masked tokens that have returned to their initial value. Red indicates the changes in the final texts. It can be seen that for small masking fractions such as $t/T = 0.1$, most of the tokens do not change after masking, while the amount of changed tokens far exceeds the unchanged ones near the phase transition at $t/T = 0.5$, hinting at the long-range correlations emerging.

Masking fraction = 0.9

Highlighted Original Text:

The third day, September 3, the situation worsened. The weather was hot and ammunition, food and supplies were nearly completely exhausted . Since the previous afternoon, North Korean mortar barrages had alternated with infantry assaults against the perimeter. Survivors later estimated there were about twenty separate infantry attacks repulsed. Two North Korean machine guns still swept the perimeter whenever anyone showed himself . Dead and dying US troops were in almost every fox hole. Mortar fragments destroyed the radio and this ended all communication with other US units. Artillery fire and air strikes requested by Schmitt never came. Some North Koreans worked their way close to the perimeter and threw grenades

Highlighted U-Turn Text:

information on maps of the actual burial population size. The number is probably around 30,000, we were almost completely encroached into the population as there were to 100 barr is we excavated the site on against the walls, it is estimated there were at around 30,000 and another holding room for perhaps 10,000 . It also seems highly unlikely, as with Dead Drop sites generally, that the only evidence for the storage of the firearm from the drop was more wood pieces. The other medieval site which required constant fire and perhaps continual storage is the firearm, one of which we were aware of having been stored during the same time period

Masking fraction = 0.7

Highlighted Original Text:

The third day, September 3, the situation worsened. The weather was hot and ammunition, food and supplies were nearly completely exhausted . Since the previous afternoon, North Korean mortar barrages had alternated with infantry assaults against the perimeter. Survivors later estimated there were about twenty separate infantry attacks repulsed. Two North Korean machine guns still swept the perimeter whenever anyone showed himself. Dead and dying US troops were in almost every fox hole. Mortar fragments destroyed the radio and this ended all communication with other US units. Artillery fire and air strikes requested by Schmitt never came. Some North Koreans worked their way close to the perimeter and threw grenades

Highlighted U-Turn Text:

 increased. On September 3, the situation was under control. Despite tons of ammunition , air train orders were almost completely violated. On the previous day, North Americans, farm crews and miners were heard rebelling against the perimeter. Survivors were estimated to be about twenty dead from attacks convulsing and starvation, as machine guns still swept the perimeter whenever ever they could. Dead - end US troops were in almost every fox hole for about twenty minutes; the radio and newspapers were all frequently with news of general effects, crying out for particular strikes or on the loading of vehicles. Some North Americans reported blocking way to fill the perimeter, and others

Masking fraction = 0.5

Highlighted Original Text:

The third day, September 3, the situation worsened. The weather was hot and ammunition, food and supplies were nearly completely exhausted. Since the previous afternoon, North Korean mortar barrages had alternated with infantry assaults against the perimeter. Survivors later estimated there were about twenty separate infantry attacks repulsed. Two North Korean machine guns still swept the perimeter whenever anyone showed himself. Dead and dying US troops were in almost every fox hole. Mortar fragments destroyed the radio and this ended all communication with other US units. Artillery fire and air strikes requested by Schmitt never came. Some North Koreans worked their way close to the perimeter and threw grenades

Highlighted U-Turn Text:

The next morning, March 3, the situation changed. The border was secure, ammunition, food and everybody were nearly completely met. On the previous afternoon, North Korean artillery barrister repulseated an infantry attack within the perimeter. Survivors later said there were about twenty separate infantry attacks repulseated. Two North Korean machine guns shells had the ground where anyone showed himself. Dead and wounded US troops were in wounded positions. At the time, fragments of mortar shells eliminated any communication of communication with other US troops. Exceptional fire and submunitions by Schmitt never came. The North Koreans worked their way up to the ground and threw bottles

Masking fraction = 0.3

Highlighted Original Text:

The third day, September 3, the situation worsened. The weather was hot and ammunition, food and supplies were nearly completely exhausted. Since the previous afternoon, North Korean mortar barrages had alternated with infantry assaults against the perimeter. Survivors later estimated there were about twenty separate infantry attacks repulsed. Two North Korean machine guns still swept the perimeter whenever anyone showed himself. Dead and dying US troops were in almost every fox hole. Mortar fragments destroyed the radio and this ended all communication with other US units. Artillery fire and air strikes requested by Schmitt never came. Some North Koreans worked their way close to the perimeter and threw grenades

Highlighted U-Turn Text:

 third! On the 3rd the situation worsened. The perimeter was thick and ammunition, food and fuel were nearly completely exhausted. By the late afternoon, North Korean mortar barrages still cooperated with infantry assaults against the perimeter for, later hours there were about 10 separate infantry attacks repulsed. Two North Korean machine guns still swept the perimeter without anyone but himself. Dead and dying US troops were in practically every man hole. Mortar fragments destroyed all radio and this ended all communication with other US units. Artillery fire or air support requested by Schmitt still came. Some North Koreans worked to bring them to the perimeter. The whites

Masking fraction = 0.1

Highlighted Original Text:

The third day, September 3, the situation worsened. The weather was hot and ammunition, food and supplies were nearly completely exhausted. Since the previous afternoon, North Korean mortar barrages had alternated with infantry assaults against the perimeter. Survivors later estimated there were about twenty separate infantry attacks repulsed. Two North Korean machine guns still swept the perimeter whenever anyone showed himself. Dead and dying US troops were in almost every fox hole. Mortar fragments destroyed the radio and this ended all communication with other US units. Artillery fire and air strikes requested by Schmitt never came. Some North Koreans worked their way close to the perimeter and threw grenades

Highlighted U-Turn Text:

The third day, September 3, the situation worsened. The weather was hot and ammunition, tanks and supplies were nearly completely exhausted. Since the early afternoon, North Korean artillery barrages had alternated with infantry assaults against the perimeter. Survivors later estimated there were about twenty separate infantry attacks repulsed. Two North Korean machine guns still swept the perimeter whenever anyone showed himself. Dead and dying US troops were in almost every fox hole. Mortar fragments destroyed the radio and this ended all communication with other US units. Artillery fire and air strikes requested by Schmitt never stopped. Some North Koreans worked their way close to the perimeter and threw grenades

## D IMAGE DIFFUSION

For image diffusion, we use the publicly available models from *Improved Denoising Diffusion Probabilistic Models* (Nichol & Dhariwal, 2021), trained on the ImageNet dataset at resolution $256 \times 256$. We use the class-unconditional model to ensure a class phase transition at an intermediate diffusion time. To tokenize the images in a semantically meaningful manner, we use the last-layer embeddings from a CLIP ViT-B32 (Radford et al., 2021) encoder. This procedure crops the images to the size $224 \times 224$, which get tokenized in $7 \times 7$ patches, each of dimension $32 \times 32$. The embeddings at the last layer of the CLIP encoder have dimension 768.

In Fig. 12, we provide some examples of images generated with the forward-backward protocol. In red, we highlight the patches whose CLIP embeddings show a statistically significant change with respect to the starting image ($t = 0$). In Fig. 13, we evaluate a convolutional classifier on the generated images and the starting ones to detect the inversion time corresponding to the class transition.

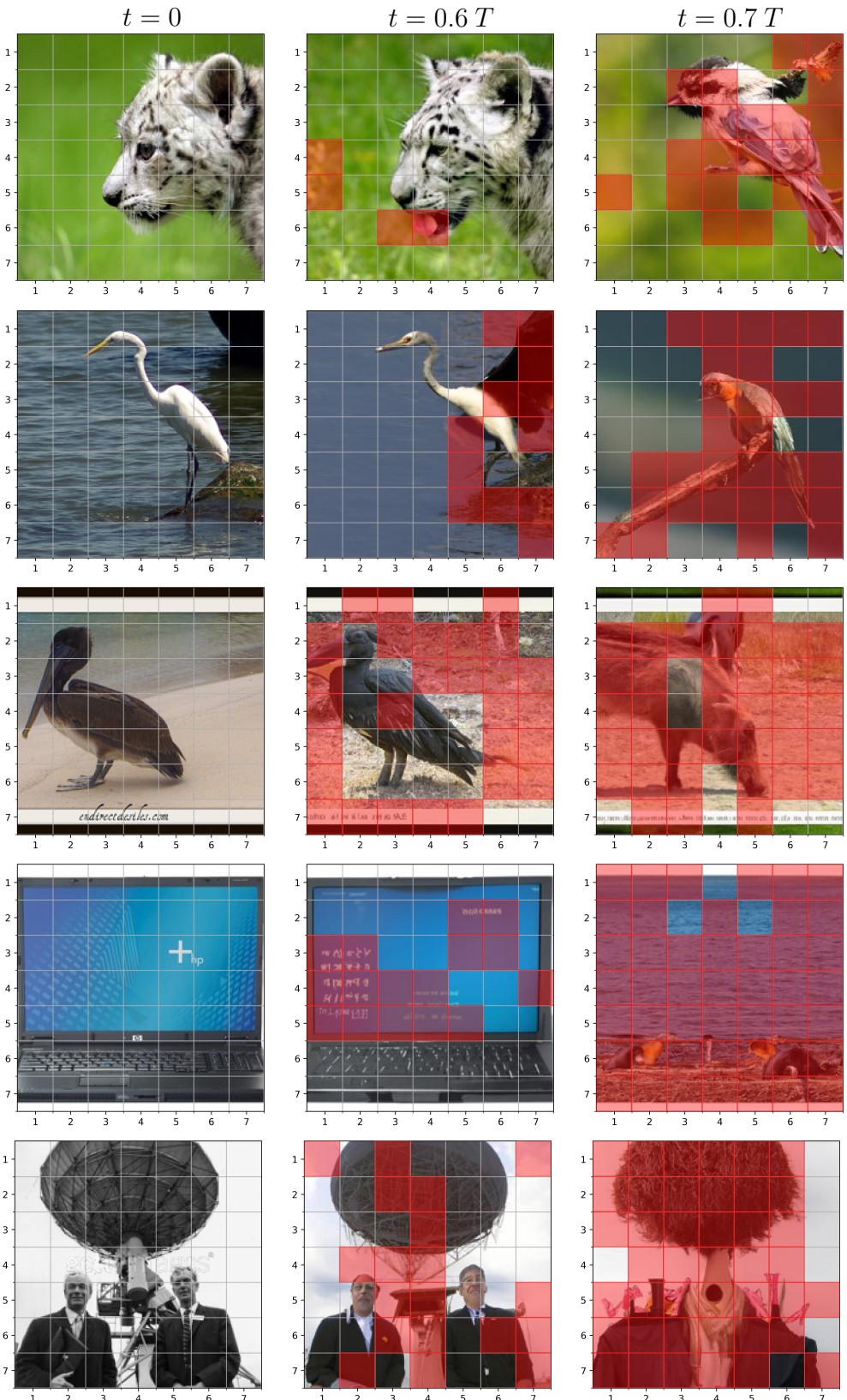

Figure 12: **Examples of images generated at different inversion times** $t$**.** The grid indicates the tokens represented inside the CLIP vision encoder. For inversion time $t > 0$, the red patches indicate the token embeddings that have a variation magnitude larger than a fixed threshold. These patches of variation appear in domains of growing size.

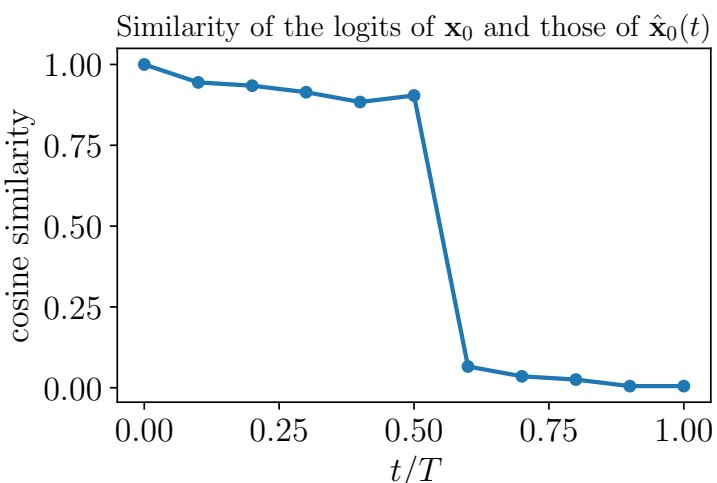

Figure 13: **Class transition in the forward-backward diffusion for ImageNet images.** Cosine similarity between the logits of a convolutional classifier computed on the starting images $\mathbf{x}_0$ and on the generated images $\hat{\mathbf{x}}_0(t)$ at different inversion times $t$. The logits are standardized on the statistics of the ImageNet validation set and the cosine similarities are averaged over 10k starting images. The convolutional classifier is a ConvNeXt Base architecture (Liu et al., 2022) pre-trained on ImageNet-1k and achieving 96.9% top-5 generalization accuracy. At short inversion times, the similarity is close to one, implying that $\mathbf{x}_0$ and $\hat{\mathbf{x}}_0(t)$ are images of the same class. At inversion time around $t \approx 0.6T$, the similarity has a sharp drop, corresponding to the class transition. Correspondingly, the susceptibility measure in Fig. 6-*(b)* has a peak.

