# OpenReview forum: "Probing the Latent Hierarchical Structure of Data via Diffusion Models"
_ICLR.cc/2025/Conference — ICLR 2025 Poster_

### Official Review · Reviewer_WL6E · 2024-10-29

**Soundness:** 4
**Presentation:** 3
**Contribution:** 3
**Rating:** 6
**Confidence:** 3

**Summary:**

This paper suggests that forward-backward diffusion experiments can be helpful in uncovering hierachical structure in data. They first study a synthetic Random Hierarchical Model and show that a peak of the dynamical susceptibility (related to correlations between blocks of tokens) occurs at a noise level where a phase transition is known to occur in the RHM (i.e. the latent class at the root changes). They then show peaks in the susceptibility in text and image experiments.

**Strengths:**

The paper is clearly written and the RHM model is nicely studied. The text and image experiments are well-designed (although I still have some questions, below, about how well the conclusions from RHM transfer to real data). I appreciate the application of ideas from Physics to ML problems.

**Weaknesses:**

Please see Questions.

**Questions:**

L57: Does such a divergence definitely indicate a hierarchical structure or are there other ways/reasons divergence could occur? i.e. is this divergence a “proof” of (or very strong evidence for) hierarchy?

L210: Clarification: so, the epsilon-process is itself a mean-field approximation of the discrete diffusion process, but then you use another mean-field on top of that to compute the correlation?

L356: Can you elaborate on why a susceptibility peak is a ‘smoking gun’ for hierarchy? Just because one nonhierarchical example doesn’t have a susceptibility peak doesn’t mean there might not be others that do?

L511: How (or does) this relate to the diffusion-as-spectral-autoregression point of view? Also there is a typo, 'trough'.

General:

Does the susceptibility divergence tell us anything about how many levels of hierarchy are likely present? Or just that there is at least one level?

The RHM is discrete, and discrete vs continuous diffusion are rather different; can you justify why RHM should be a good model for continuous data/diffusion as well?

Do the MDLM and ImageNet expts actually confirm that a phase transition occurs? Or do we just observe the susceptibility peak and infer a phase transition by analogy to RHM? In particular, it seems that for ImageNet it might actually be possible to run a classifier to determine whether the class changed.

---

> ### Author Response · Authors · 2024-11-21
>
> We thank the reviewer for finding our work clearly written, our analysis nice, and our experiments well-designed. We answer to their concerns below.
>
> **1. On the implication of a peak in the susceptibility**
>
> In this work, we demonstrate that very popular models such as Gaussian data or Gaussian random fields fail to exhibit such a peak, whereas the Random Hierarchy Model, text and images do. This work thus emphasizes the limit of modeling data as Gaussian, and points toward the need for richer structures. That said, the reviewer is correct that the observation of a peak in susceptibility does not imply that the data must be exactly tree-like. As discussed in our reply to other reviewers, we believe that mild context dependence will not affect qualitatively our conclusions. Yet, a full classification of which graphical models may capture this phenomenon is a question for the future, as now emphasized in the text (highlighted in blue).
>
> **2. Mean-field approximation**
>
> The reviewer is correct: the epsilon process can be seen as a mean-field approximation of multinomial discrete diffusion, where the uncertainty in the reconstruction is uniformly spread onto the sequence of tokens. We then use a mean-field approach, averaging over the possible rule realizations of the RHM, to compute the correlations.
>
> **3. Diffusion as spectral autoregression**
>
> We assume the reviewer is referring to the analysis presented in this [blog](https://sander.ai/2024/09/02/spectral-autoregression.html). Spectral autoregression aligns with the behavior observed in our Gaussian random field model, discussed in Section 3 and Appendix B, where higher frequencies are noised first in the diffusion process. Crucially, this perspective, as described in the blog, does not incorporate assumptions about or analyze the conditional probabilities between different scales (or frequencies). By contrast, these correlations among scales are central to the analyses of Marchand et al., 2022 and Kadkhodaie et al., 2023, who explore their structure and implications. Thus, we view these two lines of work as distinct.
>
> **Questions**
>
> 1. *Does the susceptibility divergence tell us anything about how many levels of hierarchy are likely present? Or just that there is at least one level?*
>
> In the Random Hierarchy Model, the phase transition and the associated length scale divergence hold in the limit of large depth. In practice, at finite depth, one observes a smooth crossover with an associated finite susceptibility peak, which increases with increasing depth (fixing the parameters). Conversely, the RHM with just a single level does not exhibit this phenomenology. In the case of real data, the susceptibility peak highlights the presence of structured dependencies consistent with a multi-level hierarchy but does not provide specific information on the depth.
>
> 2. *The RHM is discrete, and discrete vs continuous diffusion are rather different; can you justify why RHM should be a good model for continuous data/diffusion as well?*
>
> While images are inherently continuous, they can be described at an abstract, semantic level using discrete hierarchies. As discussed in the related work section, these hierarchical representations have been formalized in *pattern theory* (Stoyan, 1997), where the decomposition is inspired by parsing methods used in linguistics. In this framework, visual scenes are decomposed hierarchically into objects, parts, and primitives, leading to practical algorithms for semantic segmentation and scene understanding. We refer to the paragraph "Hierarchical models of images and text" in the Related Work section for further references.
>
> 3. *Do the MDLM and ImageNet expts actually confirm that a phase transition occurs? Or do we just observe the susceptibility peak and infer a phase transition by analogy to RHM? In particular, it seems that for ImageNet it might actually be possible to run a classifier to determine whether the class changed.*
>
> The susceptibility peak in real data strongly suggests an underlying hierarchical structure. In the case of images, as suggested by the reviewer, we ran a convolutional classifier (a state-of-the-art ConvNeXt pre-trained on ImageNet) to determine when the class changes (note that Sclocchi et al., 2024, performed equivalent experiments). We reported the results in Figure 12 in the updated manuscript. Clearly, in correspondence to the susceptibility peak, the class of the generated images displays a sharp transition.

---

> > ### Comment · Reviewer_WL6E · 2024-11-25
> >
> > Thank you for the helpful clarifications! I particularly appreciate the addition of Figure 12 showing that a phase transition actually occurs for ImageNet and its location coincides with the susceptibility peak. I also liked the new section on "Hierarchical models of images and text". I have increased my Soundness rating to 4 but will stick with my original score of 6 (since I share some of reviewer s11H's concerns).

---

> > > ### Author Response · Authors · 2024-11-26
> > >
> > > We thank the reviewer for appreciating our additions and clarifications.
> > >
> > > We point out that the concern of reviewer s11H about our sampling procedure is due to a misunderstanding. The two procedures are **exactly equivalent** (see the [new answer to reviewer s11H](https://openreview.net/forum?id=0GzqVqCKns&noteId=3RnKV6ijLu) and the added discussion and experiments in Appendix A.1.3). We therefore kindly invite the reviewer to reconsider their assessment of our work.

---

> > > > ### Comment · Reviewer_WL6E · 2024-11-26
> > > >
> > > > Thanks for clarifying, but my concern was more generally about whether RHM is a realistic model. I will keep my score. I can reduce my confidence if the authors would like.

---

### Official Review · Reviewer_s11H · 2024-10-30

**Soundness:** 3
**Presentation:** 3
**Contribution:** 2
**Rating:** 8
**Confidence:** 2

**Summary:**

In this paper, the authors examine the hierarchical structure in high-dimensional data by conducting forward-backward experiments within diffusion-based models. They employ a Random Hierarchy Model (RHM) for the data where the tokens of data are generated from a tree structure of latent, they also use Belief Propagation (BP) as the score function to denoise samples.

The authors focus on the phase transition of the average belief $p_L$ of the RHM's root node by analyzing an iterative mapping (Equation 7) and identifying a critical noise level $\epsilon^*$ at which the transition occurs. Based on that, they also compute the minimum layers $\tilde {l}$ needed for the transition, beyond which $p_l$ would collapse to trivial fixed points $\{1/v,1\}$, indicating either a complete reconstruction or randomization of upper latent variables. At this specific noise level, BP can modify the deepest latent layer $\tilde {l}$, yielding the maximum correlation length (i.e. big "chunks" of data token), which is the distance over which token changes remain correlated.

To characterize this effect, the authors introduce **dynamical susceptibility** which exhibits a first increase then decrease curve as expected. They further demonstrate that the dynamical susceptibility curve has the same trend for forward-backward experiments with diffusion models and synthetic RHM experiments.

**Strengths:**

1. The authors aim to capture hidden hierarchical structures within discrete data using the RHM model, with their RHM+BP framework supporting both discrete and continuous diffusion processes.

2. By applying BP for denoising, the authors rigorously analyze phase transitions in the denoising results and identify the critical noise level needed to induce a change in the data class (or low-level feature).

**Weaknesses:**

1. The paper is somewhat disorganized and hard to follow, as definitions, derivations, and experimental results are heavily interwoven. To improve clarity, consider using theorems or structured definitions to better organize the content (e.g. by moving some derivations, such as Equations 8 and 9 to appendix and summarizing them as a main theorem).

2. In practice, people use real data + score-based denoising; however, the authors use RHM data + BP denoising instead. This discrepancy is insufficiently justified, making the claim that real-world data shares the same hierarchy as RHM unconvincing. While the authors show a similar phase transition phenomenon between the RHM case and real-world diffusion case, they do not rigorously establish a connection between them. Verification by testing real-world diffusion on RHM data may strengthen this claim.

3. The results are somewhat vague and lack practical insights, as it appears that neither the RHM setup nor the forward-backward experiment has direct practical applications. Although the authors mention interpreting the "chunks" that emerge during the forward-backward experiments, they do not provide further discussion or related work on that.

**Questions:**

1. From the analysis, BP denoising appears to be a one-step method that directly samples $\hat{x_0}$ from the noisy observation $x_t$, differing from typical diffusion denoising that iteratively samples $x_{t-1}$ from $x_t$ throughout the process. Does this discrepancy exist, or are the authors also using a denoising schedule similar to real diffusion models?

2. Can we interpret the maximal correlated length achieved at an intermediate noise level (time step) as the model generating class information or lower-level features? If so, this would contrast with existing observations that diffusion processes follow a coarse-to-fine generation pattern (e.g., https://arxiv.org/abs/2303.02490), where lower-level features are generated at the beginning, not in the middle.

3. Figure 4(a) is somewhat unclear. Combined with (c), it seems the authors are suggesting that the largest correlated changing chunk appears at a masking fraction $t/T \in [0.5, 0.7]$. However, this is not immediately evident from (a) alone.

---

> ### Author Response · Authors · 2024-11-21
>
> We thank the reviewer for their feedback and address their specific concerns below.
>
> **1. Presentation**
>
> We appreciate the reviewer's feedback on improving the readability of our paper. Our work follows a structured approach: we first present our theoretical framework using the synthetic hierarchical model of data, the RHM. Then, we validate these theoretical results through numerical experiments. Finally, we test our predictions in real-world scenarios involving state-of-the-art diffusion models for both images and text. In response to your suggestion, we have taken the following steps to enhance the clarity of our presentation:
>
> - We have explicitly highlighted the structure of the paper by adding a paragraph after the introduction.
> - We have separated and presented the main definitions in a more structured, standalone format, making them easier to follow.
> - We have moved the detailed computations of the dynamical correlation length analysis (originally in Subsection 3.1.1) to the Appendix. In the main text, we have provided a more concise and accessible description of the results.
>
> We believe these revisions (highlighted in blue in the updated pdf) have significantly improved the clarity and readability of our work. We welcome any further feedback or suggestions on how we might continue to refine our presentation.
>
> **2. BP vs. neural network for denoising**
>
> Score-based generative models generate samples by reversing a forward diffusion process that progressively adds noise to data. In practice, these models employ a neural network to approximate the score function, i.e., the gradient of the log density of the data. The score enables running the backward dynamics. The score function is implicitly related to the conditional expectation $\mathbb{E}[x_0|x_t]$, where $x_t$ is a noisy observation at time $t$. Given this conditional expectation, new samples can be generated by running a time-discretized backward dynamics.
>
> For tree-like models such as the RHM, the conditional expectation $\mathbb{E}[x_0|x_t]$ can be computed exactly via message-passing algorithms like Belief Propagation (BP). As noted in L157, this corresponds to having access to a neural network that has learned the exact population score, which can be used to run the backward dynamics. However, BP also allows for the sampling of new data directly from the exact posterior $p(x_0|x_t)$ without running the reverse process. In the limit of an infinite number of diffusion steps, the two sampling processes are equivalent. Thus, our predictions with the RHM are directly comparable to real-world data.
>
> We have incorporated these clarifications into the updated manuscript (highlighted in blue).
>
> **3. Practical applications**
>
> See our reply to reviewer [ohKX](https://openreview.net/forum?id=0GzqVqCKns&noteId=wyYxlhrW0P). We want to clarify further that the primary goal of our research is to provide a fundamental analysis of the hierarchical structure of data belonging to different modalities and how diffusion models capture and utilize this structure. Our findings contribute to the growing body of work on the interpretability of latent representations in neural networks, offering a fresh perspective by examining correlated changes observed in forward-backward experiments with diffusion models. On the one side, our work supports the hypothesis that hierarchical latent structures are universal properties underlying natural data as diverse as images and language. On the other side, it opens new possibilities for future work on the interpretation of these correlated changes, e.g., in terms of the syntactic structure of a language. These further works will be data-specific, whereas the present work emphasizes the universal connection between theory and observations in very diverse settings.
>
> We clarified the text to make that point (highlighted in blue).

---

> > ### Author Response · Authors · 2024-11-21
> >
> > **Questions**
> >
> > Q1. See point (2) above.
> >
> > Q2. The reviewer is correct in interpreting the critical noise (or time) level as the point in the diffusion process that acts on class-level information. Our findings are in line with those of Sclocchi et al. (2024), who demonstrated that during forward-backward experiments, small amounts of noise only alter low-level features. Once the transition point is reached, the class becomes likely to change. Remarkably, even when the class changes, some low-level features from the original data are preserved.
> >
> > Although this might appear to contrast with the coarse-to-fine generation pattern referenced by the reviewer, these two pictures concern two different levels of description. In the coarse-to-fine view, higher-frequency components are affected early in the diffusion process, while low-frequency modes persist for longer. This is precisely the pattern we observe in simpler models like the Gaussian random field model, which we discussed in Section 3.3 and Appendix B. This viewpoint may be an appropriate starting point to describe the effect of diffusion on images at a geometric or power spectrum level.
> >
> > By contrast, our empirical results test our predictions at a semantic level (remember we consider a CLIP encoding). For images, this means that features can correspond to parts of objects - such as the eyes, mouth, and nose of a face - rather than simple geometric or frequency components. The RHM appears to be a useful model for describing these high-level aspects of images.
> >
> > We have added a citation to this work in the related work section, together with the discussion above (highlighted in blue).
> >
> >
> > Q3. The correlation length measures the distance over which the fluctuations of changes are correlated. Figures 4(b) and 4(c) show that the tokens are changed together at a maximal distance when the masking fraction is between 0.5 and 0.7. As the noise level increases further, the correlation length decreases, indicating that changes become less correlated. It’s important to note that measuring fluctuations and establishing correlation length requires statistical analysis across many instances. Therefore, the pattern we describe is not discernible from a single example in Figure 4(a) alone, which is meant to illustrate the process. We hope this clarifies the reviewer's concern.

---

> > > ### Comment · Reviewer_s11H · 2024-11-24
> > >
> > > Thanks for the great rebuttal, I have raised my rating to 6. I appreciate the authors' efforts to enhance the clarity and rigor of their work (Point 1), and I also found their clarifications on the implications, including the universality of their findings (Point 3) and the interpretation of the semantic aspect (Q2) very insightful.
> > >
> > > However, I remain skeptical about using BP+RHM to mimic real-world diffusion (Point 2 and Q1). While I agree with the authors that if the data follow RHM, BP corresponds to the optimal posterior estimation $E[\hat{x}_0|x_t]$, a major issue is that the $\epsilon$-process is not equivalent to the forward-backward experiment. In the forward-backward experiment, simulating the reverse SDE typically requires multiple posterior estimations. Relying on a single posterior estimation of $\hat{x}_0$​ would yield significantly different results—often resembling a blurred average of the training data. Since the $\epsilon$-process analyzes $\hat x_0$​, its conclusions differ from the subsequent experiments in Section 4.

---

> > > > ### Author Response · Authors · 2024-11-26
> > > >
> > > > We thank the reviewer for reconsidering their score. Regarding the concern raised in the last comment, it seems there is a misunderstanding. As highlighted in Section 2.2.1 and in our earlier response, BP does not only provide access to the **posterior mean** $\mathbb{E}[\hat{x}_0|x_t]$ but also to the full **posterior distribution** $p(\hat{x}_0|x_t)$.
> > > >
> > > > In our approach, we do not sample data using the posterior mean. Instead, **we sample from the posterior distribution** using the following procedure. We begin by sampling the root node using the marginal probability computed by BP. Then, we condition on the sampled symbol, update the beliefs, and sample one latent variable at the next layer $L-1$. This procedure is repeated node by node, descending through the tree until we generate a complete configuration at the bottom layer (cf. Mezard and Montanari, 2009).
> > > >
> > > > As mentioned in our previous answer, this sampling approach is **exactly equivalent** to running the reverse process using the posterior means in the limit of an infinite number of diffusion steps. Thus, our results on the RHM do not differ from the subsequent experiments in Section 4. To further clarify this point, we have added a detailed explanation of our sampling procedure to Appendix A.1.3. Moreover, we have added **a new experiment** for masking diffusion of the RHM, comparing the correlation functions and the dynamical susceptibility obtained with the two sampling methods, i.e., sampling from the posterior computed with BP and running the backward diffusion dynamics using the score function. Figure 8 of the updated manuscript shows that they yield identical results.
> > > >
> > > > We thank the reviewer again, and we hope that our answer and the new data resolve their skepticism.
> > > >
> > > > Mezard, M. and Montanari, A. (2009). Information, physics, and computation. Oxford University Press.

---

> ### Comment · Reviewer_s11H · 2024-11-26
>
> Thanks for the clarifications and additional experiments. I now understand that the authors are analyzing the entire progressive sampling process, which resembles the forward-backward experiment. Moreover, they use mean-field theory for simplification, without which, understanding the sampling process could be challenging for unstructured data distributions.
>
> I think the paper is accomplished, but I am still not very confident about how realistic the RHM is. I can increase my ratings and lower my confidence score if the authors would like.

---

> > ### Author Response · Authors · 2024-11-27
> >
> > We thank the reviewer for their appreciation and for proposing to raise their score. We believe it would be warranted since although the RHM is an idealized model of data, it makes non-trivial predictions confirmed both in image and text datasets.

---

### Official Review · Reviewer_ohKX · 2024-11-04

**Soundness:** 3
**Presentation:** 3
**Contribution:** 3
**Rating:** 6
**Confidence:** 3

**Summary:**

This paper aims to understand diffusion models through a hierarchical latent variable model. Through this framework, this paper demonstrates the connection between the noise level and the hierarchical levels, as evidenced by a transition phrase. This paper builds on the tools from physics and illustrate their theoretical model with empirical results on practical models.

**Strengths:**

1. The hierarchical perspective provides novel insights into the diffusion model's mechanism and the application of physics is also refreshing. I feel that the community can benefit from these insights, which may give rise to empirical advancements.
2. The paper is well written and clearly communicates the main ideas.
3. The experiments on natural data (image/text) support the theoretical claims.

**Weaknesses:**

1. It'd be great to see attempts at utilizing the theoretical/empirical observations to advance practical model design. Some discussions along this direction would also be appreciated.
2. The tree model seems overly simplified for real-world data like images and languages. For example, one would imagine two high-level variables could become co-parents for some low-level variables, thus breaking the tree structure. I would appreciate a discussion on this limitation and the applicability of the theoretical framework to more general latent models.

**Questions:**

Please refer to the weaknesses section.

---

> ### Author Response · Authors · 2024-11-21
>
> We thank the reviewer for appreciating our work, recognizing that it is well-written and that our theoretical claims are supported by experiments on natural data. Below, we address the reviewer's specific concerns.
>
> **1. Practical applications**
>
> Fundamental science contributes to practical problems by inspiring follow-up research on different time scales.  In this specific case, the theoretical and experimental results we provide have quite direct potential for practical applications. The most important one concerns the interpretability of deep networks - a central issue of the field. The hierarchical representation they build is believed to reflect the combinatorial structure of data. Tools to study the latter are scarce. Here, we show that the effect of changing latent variables at different depths can be studied by monitoring the noise level in forward-backward experiments, opening a new avenue to characterize data structure.  Finally, the presence of a transition at a certain noise level suggests that it may be especially useful to train diffusion models particularly around this noise value - an idea that practitioners just started to explore [1].
>
> As suggested by the reviewer, we added these two points in our conclusions in the revised pdf document (highlighted in blue).
>
> [1] Barceló, R. et al. Avoiding mode collapse in diffusion models fine-tuned with reinforcement learning. arXiv preprint (2024).
>
> **2. Context-free vs. context-sensitive models**
>
> The question raised by the reviewer is both interesting and subtle. On the one hand, some apparently non-tree-like graphs of latent variables can be made tree-like, if one allows for complex latent variables that encode more information. On the other hand, this is not always the case, and for example, context-dependence can be present. More broadly speaking, controlling analytically diffusion models for a general graph of latent variables is untractable - in fact, it can take a time exponential in the number of latent variables just to sample from the model.
>
> In practice, it is known that context-free grammars are not expressive enough to capture all phenomena in the syntax of natural languages, requiring *mildly context-sensitive* models [2]. Our observations on WikiText thus support that our conclusion holds beyond context-free grammars, at least for mildly context-dependent structures. In the future, building models that depart gradually from a context-free (tree-like) structure may give a handle to progress on this difficult question.
>
> In the revised manuscript, we have added a discussion in the conclusion section to address the limitations of the tree model and the potential for incorporating context dependencies in future work (highlighted in blue).
>
> [2] Jäger, G., and James, R. Formal language theory: refining the Chomsky hierarchy. Philosophical Transactions of the Royal Society B: Biological Sciences 367, no. 1598 (2012): 1956-1970.

---

> > ### Comment · Reviewer_ohKX · 2024-11-25
> >
> > Thank you for your feedback! I will maintain my current rating.

---

### Official Review · Reviewer_7ZQW · 2024-11-04

**Soundness:** 3
**Presentation:** 2
**Contribution:** 3
**Rating:** 6
**Confidence:** 2

**Summary:**

The paper examines the hierarchical correlation structures among input tokens using a dynamic correlation function and dynamical susceptibility within a forward-backward experimental framework. These variables reveal how two input tokens respond to perturbations when attempting to recover data from noisy inputs. Analyzing diffusion and language models, the study demonstrates an anticipated correlation aligned with spatial structures.

**Strengths:**

(1) The paper introduces novel approaches for analyzing the structure of inputs using pretrained diffusion and language models.

(2) The authors offer a thorough analysis and derivation, with experimental results closely aligning with theoretical expectations.

(3) Multiple schematic diagrams and data visualizations are included, providing valuable insights into the methods.

**Weaknesses:**

(1) The paper’s presentation could be improved. While there are numerous figures to aid understanding, the main text is somewhat challenging to follow.

(2) Why is the σ in Equation 3 binary? Wouldn’t a continuous measurement be more appropriate? For instance, a small difference in pixel values might not alter the semantic structure of the images, but it would be captured by binary measurement.

(3) Shouldn’t the spatial correlation structures be content-dependent? For example, if the bird and the laptop in Figure 5 were moved slightly farther from the camera, would this change affect the result shown in Figure 2?

**Questions:**

See weaknesses.

---

> ### Author Response · Authors · 2024-11-21
>
> We thank the reviewer for recognizing the novelty of our contribution and acknowledging the rigor of our analysis, along with the solid experimental validation provided. Below, we address the reviewer's specific concerns.
>
> **1. Presentation**
>
> We have made several improvements to the presentation and structure of our paper in response to feedback from multiple reviewers, including:
>
> - We have added a new paragraph at the end of the introduction section that explicitly outlines the structure of the paper, providing a clearer roadmap for the reader.
> - We have isolated the main definitions from the core text, presenting them in a more structured and standalone manner to improve readability.
> - We have moved the detailed computations of the dynamical correlation length analysis (originally in Subsection 3.1.1) to the Appendix. In the main text, we have provided a more concise and accessible description of the results.
>
> We welcome any further suggestions for enhancing the clarity of our work and are open to elaborating on specific points that may require additional explanation.
>
> **2. Binary vs. continuous spin variables**
>
> In Equation 3 (Equation 2 of the updated manuscript), our choice to use binary spin variables $\sigma_i$ stems from the discrete nature of the model under consideration in that section, where features are equidistant and represented categorically. However, we agree that a continuous measure is indeed more suitable for continuous data types, such as images. Specifically, for image data, we account for the continuity of variations by measuring the L2 distance between patch embeddings before and after the forward-backward procedure (see Equation 7). To clarify this point, we have added a sentence when introducing the discrete spin variables, noting that we will extend them to accommodate continuous data in the subsequent sections (highlighted in blue).
>
> **3. Content dependence of spatial correlation structure**
>
> We acknowledge that spatial correlations between changes in a single image are influenced by the content. Specifically, variations in camera distance or object placement can alter the spatial correlation structure of an individual image. Nevertheless, our analysis in Figure 6 reports average spatial correlations across a large set of images. These average correlations are robust to individual content variations as long as the data distribution of the initial images remains consistent. In other words, while the spatial correlation for any individual image is content-dependent, the trends in Figure 6 represent an aggregate measure, capturing the statistical properties of the dataset.
>
> We hope our responses provide sufficient clarification and would be happy to elaborate further or address additional concerns as needed.

---

> > ### Comment · Reviewer_7ZQW · 2024-12-01
> >
> > I thank the authors for their reply. I appreciate the authors' efforts in providing the modified draft with better clarity. I am still a bit concerned about Figure 5 & Figure 6 in that the trend of correlations from 0.1T to 1.0T is not consistent. While I understand these are the results from different models on different datasets, I am wondering what's the reasons behind this inconsistency. Moreover, showing the trend of these curves with the statistically uncertain range can help us understand the results better.

---

> > > ### Author Response · Authors · 2024-12-02
> > >
> > > We thank the reviewer for their response and feedback. We believe the reviewer is referring to Figures 4 and 6, which present the data for text and images, rather than Figures 5 and 6. These two datasets inherently have very different structures. As a result, it is expected that their correlation functions at different times will not be identical. Similarly, in the RHM, varying the parameters of the model or the diffusion process also leads to changes in the shape of the correlation functions and the location of the susceptibility peak.
> > >
> > > However, the presence of a peak in correlation length and susceptibility remains consistent across both modalities. Our work relates the presence of this peak, and not its precise location, to a latent hierarchical structure.
> > >
> > > As requested by the reviewer, in the revised manuscript we will add the statistical error bars, whose size is relatively small and do not change our conclusions.

---

### Meta-Review · Area_Chair_QoTv · 2024-12-20

**Metareview:**

This paper investigates the hierarchical structure of data using forward-backward experiments with diffusion models. The authors propose that changes in data occur in correlated chunks, with a characteristic correlation length that diverges at a critical noise level associated with a phase transition. These predictions are supported by experiments on synthetic hierarchical data (using a Random Hierarchy Model, RHM) and real-world datasets. The results demonstrate that susceptibility peaks observed at specific noise levels correspond to transitions in the latent structure.

**Additional Comments On Reviewer Discussion:**

Concerns were raised about the practical implications of the findings, the realism of the RHM as a model for continuous data, and clarity in linking RHM conclusions to real-world diffusion experiments. The authors addressed these issues via new experiments (e.g., showing phase transitions in ImageNet with a classifier), and manuscript revisions. While some skepticism remains about the general applicability of RHM, the reviewers converged on being in agreement about the paper's acceptance.

---

### Decision · Program_Chairs · 2025-01-22

Accept (Poster)